# Single-cell transcriptome analysis of avian neural crest migration reveals signatures of invasion and molecular transitions

Jason A Morrison[1†], Rebecca McLennan[1†], Lauren A Wolfe[1‡], Madelaine M Gogol[1], Samuel Meier[1§], Mary C McKinney[1], Jessica M Teddy[1], Laura Holmes[1], Craig L Semerad[2], Andrew C Box[1], Hua Li[1], Kathryn E Hall[1], Anoja G Perera[1], Paul M Kulesa[1,3*]

[1]Stowers Institute for Medical Research, Kansas City, United States; [2]University of Nebraska Medical Center, Omaha, United States; [3]Department of Anatomy and Cell Biology, University of Kansas School of Medicine, Kansas City, United States

*For correspondence:
pmk@stowers.org

[†]These authors contributed equally to this work

Present address: [‡]Department of Biostatistics, Bioinformatics and Epidemiology, Fred Hutchinson Cancer Research Center, Seattle, United States; [§]Broad Institute, Cambridge, United States

Competing interests: The authors declare that no competing interests exist.

**Abstract** Neural crest cells migrate throughout the embryo, but how cells move in a directed and collective manner has remained unclear. Here, we perform the first single-cell transcriptome analysis of cranial neural crest cell migration at three progressive stages in chick and identify and establish hierarchical relationships between cell position and time-specific transcriptional signatures. We determine a novel transcriptional signature of the most invasive neural crest Trailblazer cells that is consistent during migration and enriched for approximately 900 genes. Knockdown of several Trailblazer genes shows significant but modest changes to total distance migrated. However, in vivo expression analysis by RNAscope and immunohistochemistry reveals some salt and pepper patterns that include strong individual Trailblazer gene expression in cells within other subregions of the migratory stream. These data provide new insights into the molecular diversity and dynamics within a neural crest cell migratory stream that underlie complex directed and collective cell behaviors.
DOI: https://doi.org/10.7554/eLife.28415.001

## Introduction

Neural crest migration is crucial to vertebrate organogenesis. Signals within the dorsal neural tube and microenvironments through which neural crest cells travel direct cells in multicellular, discrete streams to precise peripheral targets. Time-lapse analyses have captured the complexity of neural crest cell behaviors that include the transition from the dorsal neural tube to directed migration and movement in collective groups. Cell morphologies and behaviors appear to vary depending on cell position within a migratory stream. Despite these observations at the population level, our understanding of how migrating neural crest cells interpret and respond to microenvironmental signals at the molecular level and coordinate their behaviors remains poorly understood. Thus, there is a tremendous need to interrogate the genomic features of migrating neural crest cells in a spatial and temporal manner with the goal of elucidating the underlying mechanisms of directed and collective cell migration.

The finely-sculpted pattern of discrete neural crest cell migratory streams and their broad differentiation potential have led to fascinating questions of the underlying biology (*Kulesa and Gammill, 2010*). Neural crest cells undergo an epithelial-to-mesenchymal transition to exit the dorsal neural tube in a rostral-to-caudal timed manner. As cells exit the neural tube and encounter the paraxial mesoderm, signals from within the neural tube and paraxial mesoderm sculpt cells onto stereotypical migratory pathways. By navigating through different microenvironments in a directed manner and

moving collectively in discrete streams, subpopulations of neural crest cells are distributed throughout the embryo. In the head, neural crest cells migration directs cells into regions of the face, eye, and branchial arches where cells play an important role in craniofacial patterning. Cranial neural crest cells travel in multicellular streams just underneath the surface ectoderm, presenting the opportunity to visualize and interrogate the in vivo cellular and molecular features of the migration process. Furthermore, the chick embryo is well suited to studying gene expression heterogeneities within neural crest cell migratory streams since cell isolation and single-cell RT-qPCR techniques have been optimized (*Morrison et al., 2012*, *2015*).

Genomic analyses of premigratory neural crest cells or entire stream populations have revealed transcription factors and signaling pathways that may play critical roles during neural crest induction, specification, epithelial-to-mesenchymal transition and early migratory events (*Gammill and Bronner-Fraser, 2002*; *Adams et al., 2008*; *Simões-Costa et al., 2014*; *Simões-Costa and Bronner, 2015*). This information is typically gained by isolation and bulk RNA-seq analysis of premigratory and migrating neural crest cells at a single time point, yet provides an important foundation to begin functional studies to further elucidate aspects of neural crest biology (*Simões-Costa and Bronner, 2015*, *2016*). However, what remains unclear is our understanding of the dynamic nature of gene expression changes within migrating neural crest cells and how molecular patterns are transduced into directed and collective cell migration behaviors.

To begin to address this, we recently profiled subpopulations of migrating cranial neural crest cells manually dissected and isolated by fluorescence activated cell sorting (FACS) from distinct stream positions at successive developmental stages by RT-qPCR in chick (*McLennan et al., 2012*, *2015*). We discovered regional expression of genes (96 genes selected from the literature) within a typical cranial neural crest cell migratory stream and confirmed the expression of a subset of genes by multiplexed fluorescence in-situ hybridization and quantitative measurement (*McLennan et al., 2012*, *2015*). Intriguingly, by optimizing a single-cell RT-qPCR method (*Morrison et al., 2015*), we were able to identify a unique molecular signature of 16 out of 96 genes that was stable and consistent in a subset of cranial neural crest cells confined to the invasive front of the migratory stream that we termed 'trailblazers' (*McLennan et al., 2015*). Over-expression of either of two transcription factors (*TFAP2A, HAND2*) upstream of the trailblazer genes in the follower cell subpopulation led to alterations in stream morphology (*McLennan et al., 2015*) suggesting a functional role for trailblazer genes.

Here, we leverage our previous RT-qPCR data and exploit advances in whole transcriptome amplification and cell isolation to enable single-cell RNA-seq (scRNA-seq) of neural crest migration. Our goal was to distinguish inherent variations between individual cells within a neural crest cell migratory stream. To do this, we compare gene expression fingerprints from 469 cells collected from the front, lead, and trail subregions of a typical cranial neural crest cell migratory stream at three developmental stages corresponding to initiation of migration from the dorsal neural tube, active migration towards and colonization of the branchial arches. We additionally profiled ~100 migrating neural crest cells collected from explanted cranial neural tube cultures for comparison to in vivo transcriptional signatures. We determine unique transcriptional signatures based on cell position within a stream during migration using bioinformatics tools such as unbiased hierarchical clustering and principle component analyses (PCA). Our bioinformatics analysis provides spatial transcriptomic signatures for neural crest cell subpopulations within the migratory stream and in vitro, independent of manual dissection location. We examine expression of a small subset of the trailblazer transcriptional signature by RNAscope and perform loss-of-function of a subset of trailblazer genes and the effects on neural crest migration. These data provide a comprehensive knowledge base from which to identify and functionally test genes and signaling pathways critical to neural crest migration and organogenesis.

## Results

### Bulk RNA-seq analysis affirms a rich diversity of gene expression between the invasive front and remainder of the neural crest cell migratory stream

With our previous RT-qPCR analysis that revealed a set of 16 out of 96 genes highly expressed and consistent during migration in a subset of cells restricted to the invasive front, we sought to expand the gene list in an unbiased manner. To do this, we first performed bulk RNA-seq on cranial neural crest cells isolated from the invasive front and remainder of the migratory stream at two successive developmental stages during migration and branchial arch entry (*Figure 1A* and *Figure 1—figure supplement 1*; *Hamburger and Hamilton, 1951*, HHSt13 and HHSt15). At these successive developmental stages, chick cranial neural crest cells of the pre-otic stream adjacent to rhombomere 4 (r4) are mid-migration and colonizing the branchial arch targets, respectively (*Figure 1A*). We fluorescently labeled premigratory neural crest cells with Gap43-YFP (at HHSt8-9; 4–7 somites), reincubated eggs until HHSt13 or HHSt15 and carefully manually dissected cells from the migratory front (5% of the stream, termed 'Front') and remainder of the stream (95% of the stream, termed 'Stream') (*Figure 1A*). We then used fluorescence-activated cell sorting (FACS) to isolate neural crest cells from manually dissected tissues to perform bulk RNA-seq.

Hierarchical clustering showed three distinct features of the data (*Figure 1B*). First, we find good correlation of expression levels between the three biological replicates, validating our cells isolations (*Figure 1B*). Second, there are statistically significant differences in the expression levels of subsets of genes between the different subpopulations isolated, confirming a conserved invasion signature within the front migrating cells (*Figure 1B* and *Figure 1—source data 1*). Third, subsets of genes expressed in the front subpopulation were either consistently up- or down-regulated in comparison to the remainder of the stream at both developmental stages (*Figure 1B* and *Figure 1—source data 1*).

In order to analyze the bulk RNA-seq data in more general terms, we calculated the mean of the biological replicates and determined the hierarchical clustering based on mean values (*Figure 1C*). These data reaffirmed the significant differences in the expression levels of subsets of genes between the front and remainder of migrating neural crest cells, and included aspects of the 16 gene scRT-qPCR Trailblazer signature (*McLennan et al., 2015*) (*Figure 1C* and *Figure 1—source data 1*). We find there is a subset of genes enriched in some neural crest cells at the invasive front compared with the remainder of the stream, with consistent expression at both developmental stages and unique to the previous 16 gene scRT-qPCR signature (*Figure 1D* and *Figure 1—source data 1*; n = 23 genes). In contrast, we find a subset of genes reduced in migrating front cells compared with the remainder of the stream, with consistent expression at both developmental stages (*Figure 1E*; m = 19 genes). In comparison to our previous identification of 16 genes out of 100 analyzed by scRT-qPCR (*McLennan et al., 2015*), bulk RNA-seq analysis validated that *ITGB5* and *GPC3* (HHSt13), and *BAMBI* and *PKP2* (HHSt15) were enriched in FRONT versus stream samples. *EPHA4* (HHSt13) and *CDH7* (HHSt15) were reduced. Together, these bulk RNA-seq analyses affirm there is a rich spatio-temporal diversity of gene expression depending on whether a neural crest cell is within the invasive front versus any other position within the stream and reveal genes that are either enhanced or reduced consistently at the invasive front.

### Single-cell RNA-seq identifies gene expression variances based upon spatial position within the neural crest cell stream and temporal progression along the migratory pathway

To better characterize unique transcriptional signatures and gene expression heterogeneity during cranial neural crest migration, we isolated and profiled individual cells from different stream positions at three developmental stages (HHSt11,13,15; *Figure 2A*, *Figure 2—source data 1* and *Figure 2—figure supplement 1*). These three progressive developmental stages were selected based on the different migratory events including recently delaminated from the neural tube (HHSt11), invasion of the paraxial mesoderm (HHSt13) and entry into the second branchial arch (HHSt15) with ~8 hr in between the stages. Since there are few recently emigrated neural crest cells at HHSt11, we could take advantage of single-cell analysis whereas with bulk RNA-seq described above

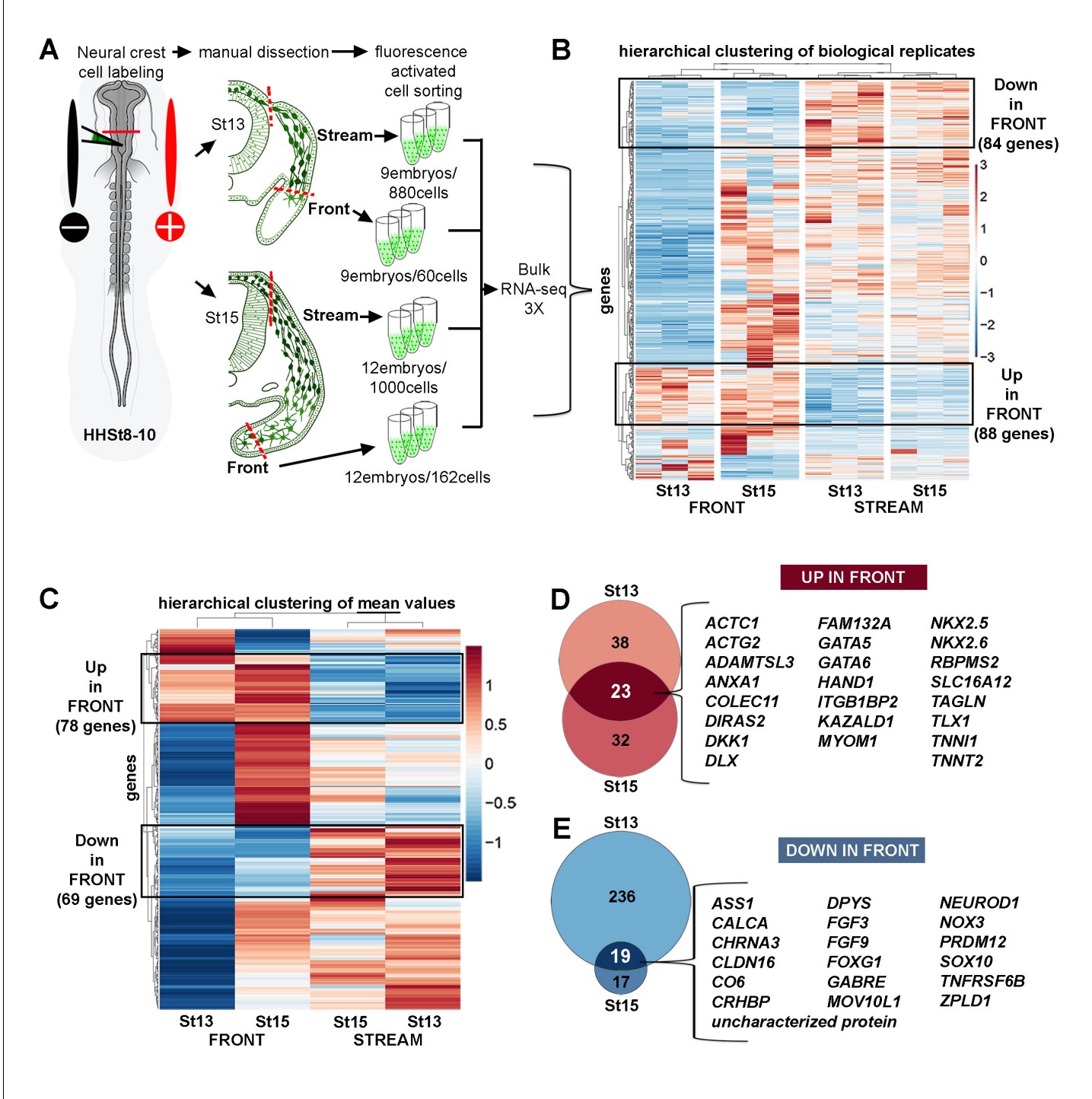

**Figure 1.** Bulk RNA-seq confirms the unique molecular profile of cells at the invasive front of the neural crest stream. (**A**) Schematic representation of method used for harvesting samples from the cranial NC stream. Invasive front (Front) is the ventral-most 5% of NC cells. Stream is the remaining 95% of NC cells. n values as shown. (**B**) Heatmap of biological replicates of differentially expressed genes in HHSt13 and 15 Front and Stream (n = 477 genes). Black boxes contain gene clusters expressed at significantly higher and lower levels by the Front. (**C**) Heatmap of mean values of differentially expressed genes in HHSt13 and 15 Front and Stream samples (n = 477 genes). Black boxes contain gene clusters expressed at significantly higher and lower levels by the Front. (**D**) Venn diagram displaying the numbers of genes enriched at the Front of the NC stream at both HHSt13 and 15 with a union of 23 genes enriched in both comparisons. (**E**) Venn diagram displaying the numbers of genes reduced at the front of the NC stream at both HHSt13 and 15 with a union of 19 genes reduced in both comparisons. *Hamburger and Hamilton (1951)*; St, stage.

DOI: https://doi.org/10.7554/eLife.28415.002

*Figure 1 continued on next page*

*Figure 1 continued*

The following source data and figure supplement are available for figure 1:

**Source data 1.** Bulk RNA-seq differential expression gene lists.

DOI: https://doi.org/10.7554/eLife.28415.004

**Figure supplement 1.** Bulk RNA-seq quality control analyses.

DOI: https://doi.org/10.7554/eLife.28415.003

we would not have been able to further dissect into invasive front and stream subpopulations. This unique approach maintained a level of spatial information within our scRNA-seq data set.

Premigratory neural crest cells were fluorescently labeled at HHSt8 with Gap-43-YFP for ease of identification during migration through unlabeled tissue for segregation of neural crest cells from unlabeled tissue by FACS (*Figure 2A*). We subdivided neural crest cell migratory streams at HHSt13 and HHSt15 into three spatially distinct, non-overlapping subregions by careful manual dissection and single cells were isolated by FACS. This included neural crest cells within the invasive front (5% of the stream, termed 'Front'), lead subregion of the stream (25% of the stream just proximal to the front, termed 'Lead'), and remainder of the stream (70% of the stream, termed 'Trail') (*Figure 2A*). At HHSt11, we did not subdivide the stream since there are relatively small numbers of migrating neural crest cells adjacent to rhombomere 4 (r4) and refer to these cells as 'Front' (*Figure 2A*). The total number of single cells analyzed after quality control was 469 (*Figure 2B*). Hierarchical clustering of single neural crest cell averages revealed statistically enriched and reduced genes between every pairwise combination of subpopulations and concordance between Lead cells at HHSt13 and HHSt15 and Trail cells at HHSt13 and HHSt15 (*Figure 2—figure supplement 2*). These data show that spatial location within the stream and temporal progression along the migratory pathway are primary determinants of gene expression.

## In vitro transcriptional signatures of migrating neural crest cells are distinct from any in vivo profile

Neural tube explant cultures are often used as an assay for neural crest cell migration studies. However, we previously showed that there are gene expression profile differences between neural crest cells grown in vitro and neural crest cells isolated from the invasive front in vivo by RT-qPCR (*McLennan et al., 2015*). That is, we found significant differences between in vitro and in vivo RT-qPCR gene expression profiles by Euclidean clustering and dissimilarity matrix plots (*McLennan et al., 2015*) that might explain the lack of collective cell migration in culture.

To better characterize in vitro cranial neural crest cell gene expression profiles, we performed scRNA-seq on neural crest cells derived from explanted neural tubes. To do this, we explanted cranial neural tubes dissected from HHSt8-9 embryos onto fibronectin-coated Mat-tek dishes and allowed cells to invade the dish for 24 hr, comparable to in vivo HHSt15. At t =+ 24 hr, neural tube explants were removed from the dish leaving the migrating cells to be collected and isolated by FACS for single-cell RNA-seq analysis. We identified several key features when we compared in vitro versus in vivo neural crest cell gene expression profiles (*Figure 2—source data 2*). First, using a Pearson correlation matrix to relate the average expression profile of all migrating neural crest cell subpopulations to show that the average in vitro profile is prominently distinct from all in vivo profiles analyzed (*Figure 2C*). Specifically, the in vitro neural crest cell gene expression profile correlates poorly with every isolated in vivo subpopulation. The closest correlation is cells most recently exited from the dorsal neural tube (HHSt11) (*Figure 2C*; *Figure 2—figure supplement 2*). To definitively ensure that the difference between in vitro and in vivo neural crest cell expression was not the product of averaging, we performed a principal component analysis (PCA) containing all single-cell RNA-seq profiles (*Figure 2D*). The PCA reveals that all in vitro single neural crest cells cluster together and show minimal overlap with any in vivo single-cell expression profile (*Figure 2D*).

To gain insight into the gene expression differences between migrating neural crest cells isolated from the embryo and in vitro, we performed a Condensed Gene Ontology (GO) Term analysis. From this analysis, there is enrichment of genes associated with biological processes such as cell signaling, transcription regulation, growth factor response, migration and proliferation by in vivo neural crest cells when compared to in vitro data (*Figure 2E*). We find that 806 genes are down-regulated in

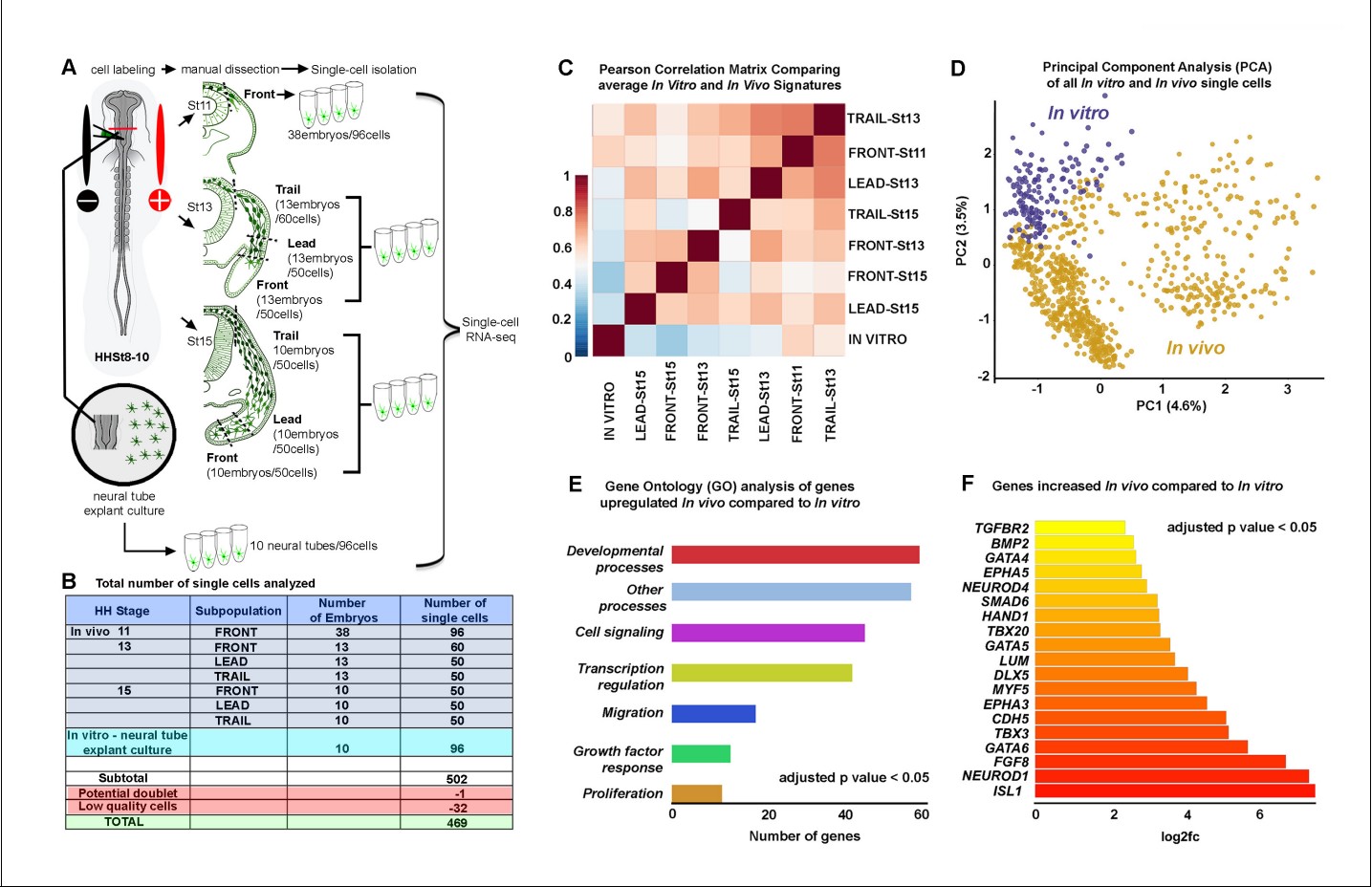

**Figure 2.** Single-cell RNA-seq shows in vitro and in vivo neural crest have distinct molecular signatures. (**A**) Schematic representation of method used for harvesting samples from the cranial NC stream. Front is the ventral-most 5% of NC cells. Lead is the 25% of NC cells immediately following the Front cells. Trail is the remaining dorsal-most 70% of NC cells. Neural crest cells grown in vitro overnight from isolated neural tubes were also isolated for this analysis. (**B**) Numbers of single cells isolated from different subpopulations. (**C**) Pearson correlation matrix of the average expression profiles, based upon all differentially expressed genes all subpopulations analyzed (n = 1355 genes). (**D**) Principal Component Analysis (PCA) of all in vitro and in vivo single cells using all differentially expressed genes (n = 1355 genes). (**E**) Condensed Gene Ontology (GO) analysis of genes upregulated in in vivo NC cells compared to in vitro NC cells (n = 806 genes). (**F**) Bar chart displaying NC relevant genes increased in in vivo NC cells compared to in vitro NC cells. HH, *Hamburger and Hamilton (1951)*; St, stage.

DOI: https://doi.org/10.7554/eLife.28415.005

The following source data and figure supplements are available for figure 2:

**Source data 1.** Single-cell RNA-seq differential expression gene lists for all spatiotemporal subpopulation pairwise comparisons.
DOI: https://doi.org/10.7554/eLife.28415.008
**Source data 2.** Single-cell RNA-seq differential expression gene list comparing in vitro and in vivo cells.
DOI: https://doi.org/10.7554/eLife.28415.009
**Figure supplement 1.** Single-cell RNA-seq quality control analyses.
DOI: https://doi.org/10.7554/eLife.28415.006
**Figure supplement 2.** Single-cell RNA-seq averages show regional and temporal expression differences within the neural crest stream.
DOI: https://doi.org/10.7554/eLife.28415.007

neural crest cells grown in vitro (e.g., *NKX2-5, NKX2-6, WNT5B, SNAI1, PAX5, PAX9, HAND1, PHOX2B, EPHA3, EPHA5, NCAM2, FGFs 4, 7, 8, 9, 10, 18, 19, AQPs 1, 3, 5, NEUROD1, NGFR, NRP1, DLXs 1, 3, 5, 6, TWIST1, TGFBR2, DSP, CDHs 5, 10, CXCL12, GATA4, GATA6, GPC3, ISL1, BMP2,* and *KDR (VEGFR2)*) compared with in vivo data (*Figure 2F* and *Figure 2—source data 2*). We also find 312 genes up-regulated in migrating neural crest cells in vitro (e.g., *WNT9A, SHH, SOX2, PAX7, IL8, MKX, CDH6* and *CDH12*) (*Figure 2—source data 2*). However, despite the

identification of 312 genes up-regulated in migrating neural crest cells in vitro, we did not find any GO terms associated with these genes. These data strongly suggest that in vivo embryonic neural crest cell microenvironmental signals influence both the spatial and temporal heterogeneities of neural crest cell gene expression and biological processes critical to migration and embryogenesis.

### Evidence for unique transcriptional signatures associated with cell subpopulations within the in vivo neural crest cell migratory stream

When we analyzed the heatmap of averaged expression profiles of all subpopulations we found regional and temporal expression differences (*Figure 2C*; *Figure 2—figure supplement 2*). This suggested a richness of diversity between each of the subpopulations analyzed. To gain insight into the extent of this diversity, we clustered all single-cell profiles as a heatmap based on subpopulation (*Figure 3*). First, we confirm that in vitro neural crest cells display up- and down-regulation of genes distinct from the in vivo subpopulations (*Figure 3*; column of cells marked 'in vitro'). Second, we identify subsets of genes that either increased (*Figure 3*, box A) or decreased (*Figure 3*, box B) with developmental stage and spatial location within the stream. Third, we discover subsets of genes that were uniquely reduced (*Figure 3*, box A') or enhanced (*Figure 3*, box C) in individual subpopulations of migrating neural crest cells. Fourth, we find that some genes show expression changes that correlate with spatial position within the stream at a single developmental time point (*Figure 3*, box D). Lastly, we identify that heterogeneity of expression among single cells within any subpopulation was common (*Figure 3*). A Pearson correlation matrix of all single-cell transcriptomes confirmed similarity among some, but not all cells within each subpopulation (*Figure 3—figure supplement 1*), indicating that our spatial-temporal locations are not completely homogeneous. These results demonstrate that each cranial neural crest cell subpopulation analyzed (Front, Lead, Trail) can be defined by distinct molecular signatures, but also include heterogeneity of expression.

### Hierarchical clustering and PCA analysis identify distinct transcriptional signatures associated with the invasive front of the neural crest migratory stream

Our previous RT-qPCR (*McLennan et al., 2015*) and bulk RNA-seq (*Figure 1*) analyses showed that cranial neural crest cells at the invasive front have portions of their expression profiles that are conserved as cells travel through different microenvironments. To determine the transcriptional signatures associated with neural crest cells at the invasive front of the migratory stream, we clustered the invasive front cells in a heatmap based on gene expression, rather than stage of development (*Figure 4A*). First, hierarchical clustering reveals that the majority of neural crest cells at the invasive front cluster together by developmental stage (*Figure 4A*; *Figure 4—source data 1*). Further, a Pearson correlation matrix shows that neural crest cells within the invasive front at HHSt13 and HHSt15 have expression profiles that cluster most similarly with Lead cell subpopulations, which we termed Lead-like (*Figure 4B*). Second, we discover two distinct cell subpopulations exist at HHSt11 termed Cluster1 and Cluster2 (*Figure 4A–B*). Third, we find a distinct cluster that contains neural crest cells isolated from the invasive front at all three developmental stages analyzed, termed scRNA-seq Trailblazers (*Figure 4A*). The single-cell PCA plot shows these five distinct cell subpopulations within the invasive front over time, with very minimal overlap in expression profile between the Trailblazer cluster and any other cell subpopulation cluster (*Figure 4C*; light blue dots). The identification of a transcriptomic signature associated with Trailblazer cells within the invasive front represents an unbiased and more comprehensive molecular profile than we previously described by scRT-qPCR (*McLennan et al., 2015*).

### The scRNA-seq trailblazer signature is uniquely defined and associated with neural crest cells mostly confined to the invasive front of the stream

To characterize the scRNA-seq Trailblazer transcriptional signature, we identified genes that are statistically enriched or reduced (*Figure 5A* and *Figure 4—source data 1*). A volcano plot shows that 964 genes are up-regulated (e.g., *BAMBI, LUM, MMP2, GATA2, GATA4, GATA5* and *GATA6*) and 406 genes are down-regulated (e.g., *DRAXIN, TFAP2A, TFAP2B, TFAP2C, TFAP2E* and *SOX10*) within the Trailblazers (*Figure 5A* and *Figure 4—source data 1*). Nineteen out of 24 genes (79%)

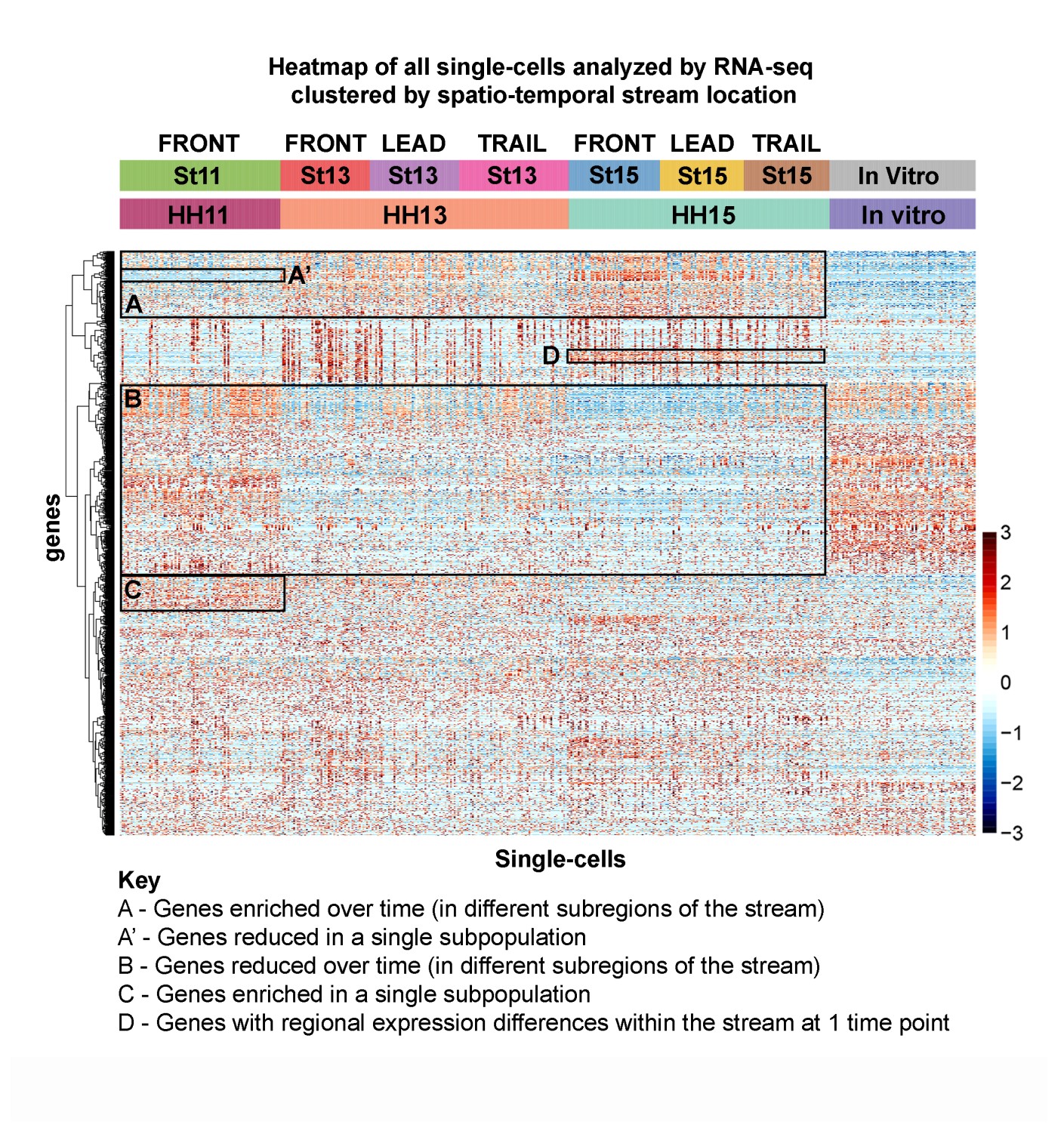

**Figure 3.** Single-cell RNA-seq exposes novel subpopulations within the neural crest stream. Heatmap of all single cells (n = 469 single cells, 1355 genes) analyzed by RNA-seq with single cells clustered by spatiotemporal location. (Box A) Genes increased with developmental time and spatial location. (Box A') Example of genes uniquely downregulated by an individual subpopulation. (Box B) Genes decreased with developmental time and spatial location. (Box C) Example of genes uniquely upregulated by an individual subpopulation. (Box D) Example of regional differences within one developmental stage.

DOI: https://doi.org/10.7554/eLife.28415.010

The following figure supplement is available for figure 3:

*Figure 3 continued on next page*

*Figure 3 continued*

**Figure supplement 1.** Pearson correlation matrix comparing the similarity of single-cell transcriptomes.

DOI: https://doi.org/10.7554/eLife.28415.011

enriched in the invasive front from our bulk RNA-seq analysis (*Figure 1D*) were also enriched in Trailblazers by scRNA-seq (*Figure 4—source data 1*). Additionally, 7 out of 19 genes (37%) reduced in the invasive front from our bulk RNA-seq analysis (*Figure 1E*) were also reduced in Trailblazers by scRNA-seq (*Figure 4—source data 1*). We also find examples of known neural crest-related genes that are up-regulated in Trailblazers (e.g., *EDN1, GPC3, NRP1, BMP2*) and include genes (*AQP1, BAMBI, PKP2*) identified in our RT-qPCR analysis (*McLennan et al., 2015*) (*Figure 5B*). There are also neural crest-related genes down-regulated in Trailblazers (e.g., *MSX1, EDNRB, PAX3, SOX10*) (*Figure 5B* and *Figure 4—source data 1*).

With an ensemble of genes that we use to define Trailblazer cells, we may characterize the presence of these cells within the invasive front. Intriguingly, the number of neural crest cells within the invasive front (2–3% of the entire migratory stream) with the Trailblazer transcriptional signature increased significantly during HHSt11 to HHSt13 and was then consistent from HHSt13 to HHSt15. At HHSt11 we found that approximately 8% of the migrating neural crest cells had a Trailblazer transcriptional signature. By HHSt13, with an estimate of approximately 500 neural crest cells in the typical migratory stream at this stage and axial level, we found that 49% of the cells within the invasive front had a Trailblazer transcriptional signature, or around 12 cells. At HHSt15, approximately 32% (or around 13 cells within the invasive front of a stream of approximately 800 cells) had a Trailblazer transcriptional signature. In an effort to be more precise about the Trailblazer signature, we filtered the 964 genes enriched in Trailblazer cells to include only genes detected in the vast majority (>90%) of trailblazer cells have a more stringent criteria (*Figure 5F*).

To determine whether the scRNA-seq Trailblazer transcriptional signature is present elsewhere in the neural crest cell migratory stream, independent of the Pearson correlation matrix result (*Figure 5C*), we generated a PCA plot of all analyzed single-cells based on the Trailblazer transcriptional signature genes (*Figure 5C*). We find that fewer than 10% of neural crest cells in other subregions of the migratory stream show expression profiles that overlap with any of the Trailblazer cells (*Figure 5C*; Lead (n = 8 out of 100 cells) and Trail (m = 10 out of 110 cells)). These data suggest that neural crest cells with a Trailblazer transcriptional signature are mostly confined to the invasive front of the migratory stream.

To better understand the correlation between our Trailblazer transcriptional signature and cell invasion we performed signaling pathway enrichment analysis using genes differentially expressed by the Trailblazers compared to all other neural crest cells harvested from the embryo (*Figure 5D*). We present both the developmentally relevant (*Figure 5D*) and irrelevant (*Figure 5—source data 1*) signaling pathways. We find that of the genes differentially expressed by Trailblazers and classified into known pathways, 81% were increased and only 19% were decreased (*Figure 5D*; red and blue numbers respectively). We find increased genes associated with pathways such as WNT/Beta Catenin, tight junction, Rho-family GTPases, EMT, integrin, gap junction, endothelin-1 and BMP signaling (*Figure 5D*). Consistent with our previous findings that VEGF/neuropilin-1 chemoattraction guides cranial neural crest cell migration (*McLennan and Kulesa, 2007*; *McLennan et al., 2010*; *McLennan and Kulesa, 2010*; *McLennan et al., 2015*), we find that increased expression of VEGF receptors 1, 2 and 3 as well as neuropilin-1 (*NRP1*) in the Trailblazers (*Figure 5D*). Further, 37 genes associated with axonal guidance signaling are expressed at significantly higher levels by the Trailblazers, including members of the BMP, EPH, MMP and WNT families (*Figure 5D*). Together, these results highlight the upregulation of genes and gene families within the Trailblazers previously identified as functionally critical to neural crest cell migration.

## The scRNA-seq analysis validates some of our original 16 gene RT-qPCR trailblazer signature

When we examined the presence of our 16 gene RT-qPCR Trailblazer signature (*McLennan et al., 2015*) within the scRNA-seq data, we found four genes (*AQP1, BAMBI, CTNNB1* and *PKP2*) enriched in both analyses (*Figure 5E*). The HHSt15 Lead-like subpopulation that was included in our RT-qPCR

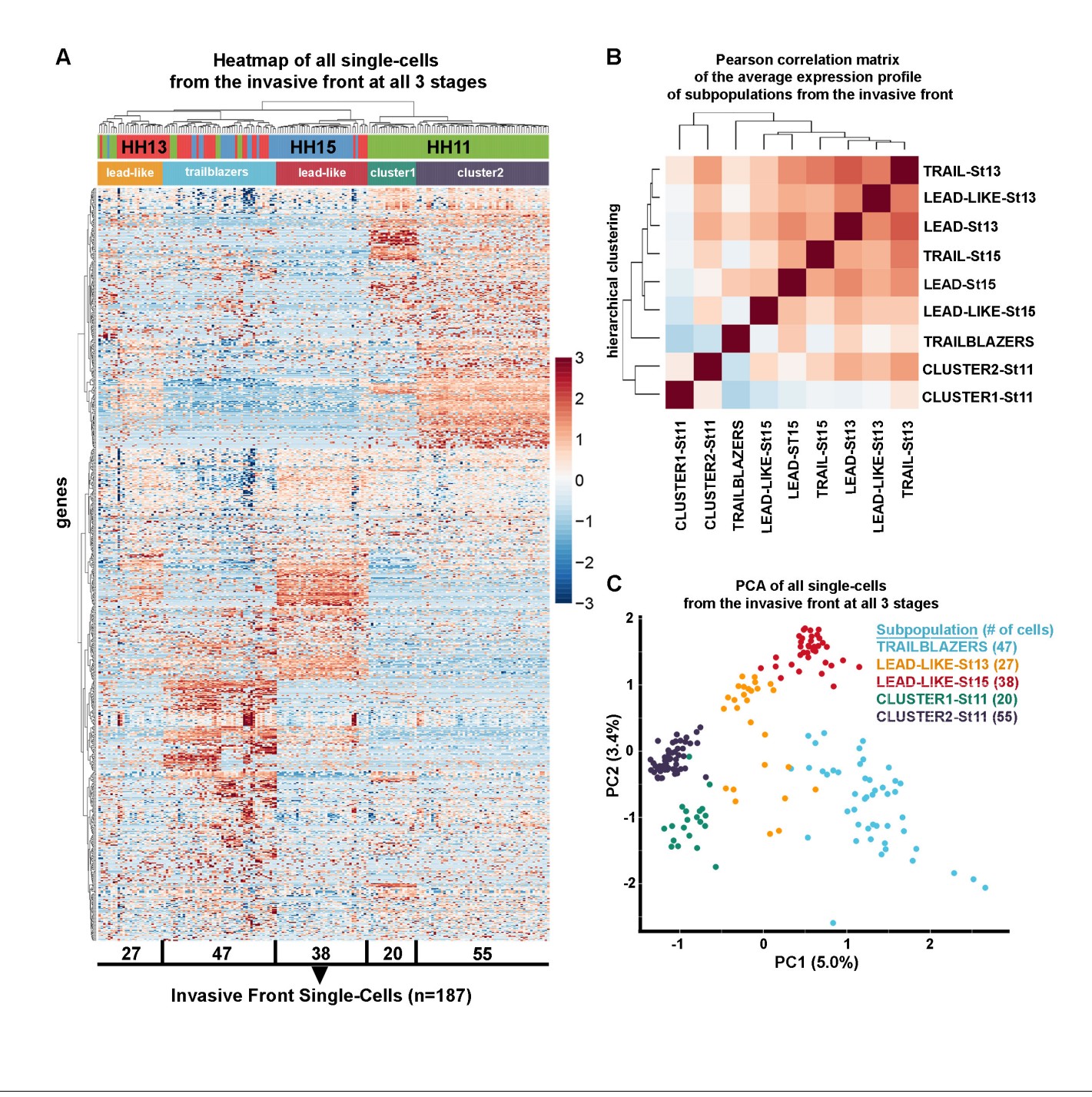

**Figure 4.** Trailblazers are a small subpopulation of cells with a distinct expression signature that exist at the invasive front of the neural crest stream, regardless of developmental stage. (A) Heatmap with hierarchical clustering of all single cells from the invasive front at HHSt11, 13 and 15 (n = 187 cells), based upon differentially expressed genes from all invasive front comparisons and invasive front compares to stream at HHSt13 and HHSt15 (n = 760 genes). (B) Pearson correlation matrix of the average expression profile of all subpopulations based upon differentially expressed genes from all invasive front comparisons and comparisons between clusters and all other in vivo cells (n = 3555 genes). (C) PCA of all single cells from the invasive front at all three developmental time points, based upon differentially expressed genes from all invasive front comparisons and invasive front compares to stream at HHSt13 and HHSt15 (n = 760 genes).

DOI: https://doi.org/10.7554/eLife.28415.012

The following source data is available for figure 4:

*Figure 4 continued on next page*

*Figure 4 continued*

**Source data 1.** Single-cell RNA-seq differential expression gene lists for all invasive front subpopulation comparisons.
DOI: https://doi.org/10.7554/eLife.28415.013

Trailblazer subpopulation (defined by manual dissection and not molecular signature) showed an increase of *ITGB5* and *NEDD9* (*Figure 5E*). Of the other 16 genes, *EPHB1* was enriched in HHSt11 Cluster1 compared to all other migrating cells profiled from in vivo (*Figure 5E*) and *CDH7* and *CXCR4* were enriched in HHSt11 Cluster2 (*Figure 5E*). Differential expression analysis between HHSt11 Cluster1 and Cluster2 showed that *CDH11*, *CXCR4*, *NEDD9*, and *TFAP2A* and *TFAP2B* were enriched in HHSt11 Cluster2 compared to Cluster1, but no gene of the 16 RT-qPCR signature genes were enriched in HHSt11 Cluster1 versus Cluster2 (*Figure 4—source data 1*).

## Expression analysis by RNAscope of individual genes associated with the scRNA-seq Trailblazer transcriptional signature are not necessarily restricted to neural crest cells at the invasive front

To visualize the expression of genes associated with the Trailblazer transcriptional signature, we optimized a novel protocol combining multiplexed fluorescence in situ hybridization (RNAscope; *Morrison et al., 2017*; *Gross-Thebing et al., 2014*) with immunohistochemistry (IHC) in chick (*Figure 6*). We took advantage of image analysis tools to quantify the fluorescence signals within subregions or individual neural crest cells by fluorescence spot counting to measure transcript levels (*Figure 6*). Together, this allowed us to detect mRNA target expression specifically within HNK1-labeled migrating neural crest cells in 3D within the intact embryo (*Figure 6* and *Video 1*).

We selected 13 genes for expression investigation based on the following criteria: (1) genes with enriched expression in Trailblazer cells; (2) high percentage of single Trailblazer cells in which the gene was detected and; (3) high level at which the gene was expressed by the Trailblazer cells. We find that none of the 13 genes tested are expressed in all of, or the majority of cells at the invasive front (*Figure 6A–M*). That is, some genes appear expressed in a few or some but not in all of the cells (~30%) of the (narrow) front migratory edge (*Figure 6A–M*). For example, Plakophilin2 (PKP2) and Kazal Type Serine Peptidase Inhibitor Domain 1 (*KAZALD1*) showed heterogeneity of expression between invasive front neighboring cells (*Figure 6A–A"*, *C–C"*; *Figure 4—source data 1*). Further, some of the 13 genes analyzed by RNAscope display salt and pepper patterns that include strong expression in a few cells in the invasive front and similarly strong signal in some cells in the beginning or middle portions of the stream (not at the front edge of migration). We found Desmin (*DES*) and Tescalcin (*TESC*) to be expressed within the invasive front (Figure B,D; green shading), but also in migrating neural crest cells within the remainder of the stream (*Figure 6B,D*; red sharing). Lastly, some of the 13 genes were not found at the invasive front, and instead appear to be expressed more strongly at various positions within the migratory stream. That is, TroponinI1 Slow Skeletal Type (*TNNI1*), Glypican3, and TFPI2 showed strong signal in some cells in the middle or back portions of the stream (*Figure 6E,G,L*).

## Knockdown of some trailblazer genes showed changes in neural crest cell distance migrated

To begin to understand the function of Trailblazer genes identified by our scRNA-seq, we performed loss-of-function experiments by morpholino transfection into premigratory neural crest cells (*Figure 6N–Q* and *Figure 6—figure supplements 1–3*). We selected 8 genes from distinct molecular families to test, based on the same criteria used for selecting genes for expression analysis, but also for predicted roles in neural crest migration. These genes included, *WNT5B* (expressed in 91% of Trailblazer cells), *PKP2* (57%), *EDN1* (57%), *MMP2* (96%), *GATA5* (77%), *GPC3* (62%), *ANXA1* (47%) and *LUM* (79%) (*Figure 4—source data 1*; *Figure 6N–Q*). We found that knockdown of 3 of these genes (*WNT5B* (p=0.0001), *PKP2* (p=0.017), *EDN1* (p=0.004)) led to a significant reduction (>13%) in neural crest cell distance migrated (*Figure 6N–Q*). Specifically, we found that *WNT5B* showed the most significant change in neural crest cell migration (21% reduction in distance migrated) (*Figure 6P,Q*). Three of these genes did not significantly reduce migration (*Figure 6Q*) and for the remaining 2 genes, we could not get sufficient morpholino transfection within the

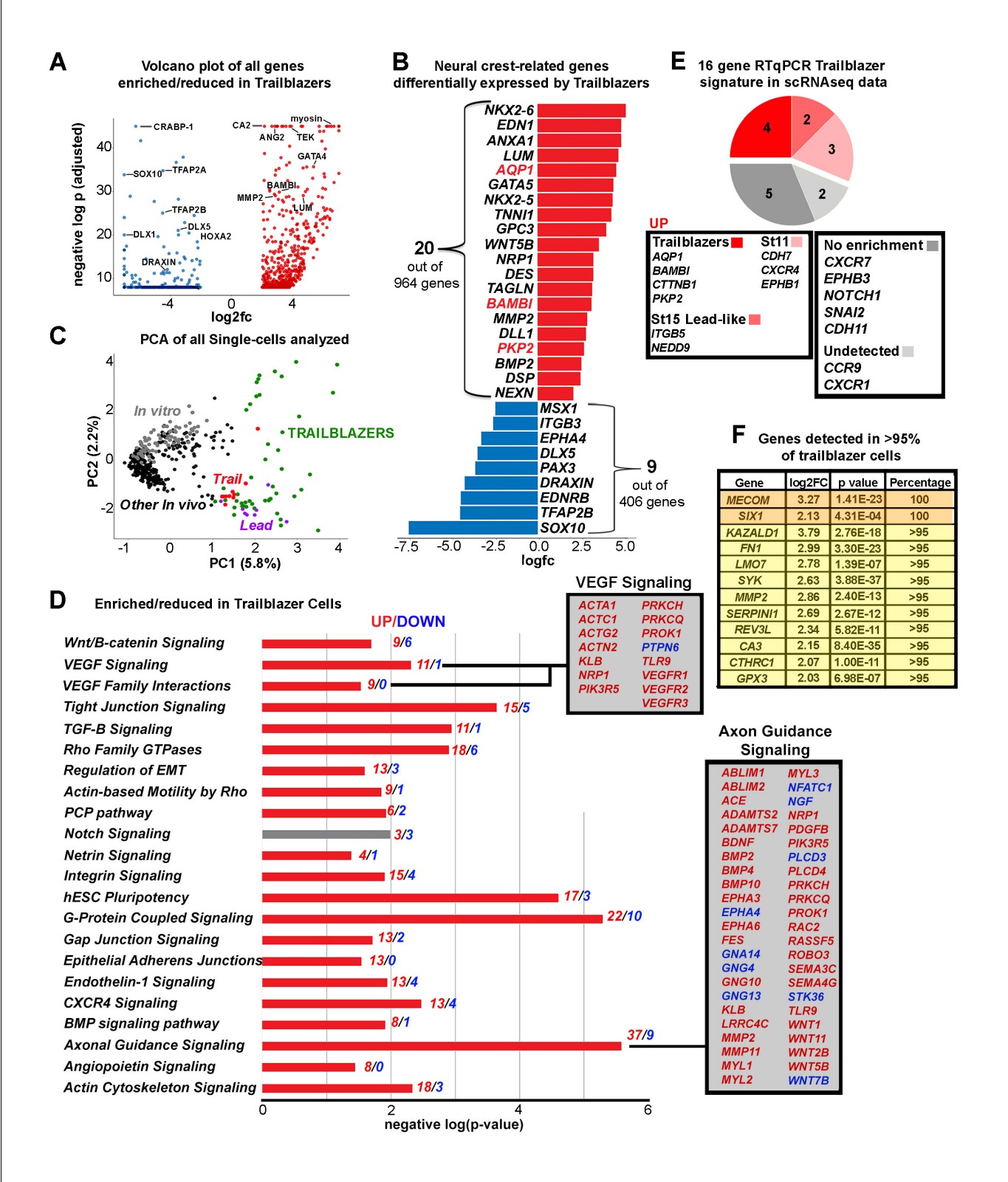

**Figure 5.** The Trailblazer signature represents tissue development, adhesion and a lack of differentiation. (A) Volcano plot of all genes statistically enriched or reduced in Trailblazers (n = 47 cells, 1370 genes). (B) Bar chart showing examples of neural crest-related differentially expressed genes in Trailblazers (red designates previously identified in *McLennan et al., 2015*). (C) PCA of all single cells analyzed showing minimal overlap of other in vivo (black dots) and in vitro (gray dots) with the Trailblazers (green dots) (n = 1370 genes). (D) Significant signaling pathways predicted to be enriched (red)

*Figure 5 continued on next page*

*Figure 5 continued*

or reduced (blue) in Trailblazers. VEGF and axonal guidance signaling are highlighted. The statistical significance was set at log (−1.3) [p<0.05]. (E) Comparison to previous RT-qPCR generated trailblazer signature. (F) Genes enriched and detected in >95% of trailblazer cells.

DOI: https://doi.org/10.7554/eLife.28415.014

The following source data is available for figure 5:

**Source data 1.** Unbiased signaling pathways enriched in single cell subpopulations.

DOI: https://doi.org/10.7554/eLife.28415.015

migrating neural crest stream and therefore did not analyze (data not shown). Although there were some genes in which high expression in the larger percentage of Trailblazers (>50%) correlated with the significant reduction in neural crest cell distance migrated (*WNT5B, GATA5, PKP2*), this was not a general feature, indicating that expression does not necessarily correlate with function.

Splice blocking morpholinos were used so that their effects could be detected by RT-PCR. For example, PKP2 MO and ANXA1 MO both resulted in exclusion of an exon (3 and 4 respectively), which could be seen by using PCR primers that flank the effected exon (*Figure 6—figure supplement 2*). Other morpholinos were harder to test due to small number of introns/exons (for example LUM), or frame shifts resulting in unknown changes (for example DLL, GATA5). Overall stream migration distance was also calculated using the HNK1 neural crest marker, and although some morpholino transfections caused a statistically significantly shorter stream, they all still migrate over 90% of the distance to the end of the branchial arch (*Figure 6—figure supplement 3*). This suggests that the morpholino transfections are cell autonomous and that wildtype neural crest cells are able to presume the role of the Trailblazers and therefore migrate into the target site.

## Identification of two distinct transcriptional signatures associated with neural crest cells isolated shortly after dorsal neural tube exit

We described above that hierarchical clustering and PCA analyses of scRNA-seq data from neural crest cells isolated from the invasive front revealed two cell subpopulations with distinct gene expression profiles at HHSt11, termed Cluster1 and Cluster2 (*Figure 4A–C*). Neural crest cells within Cluster1 and Cluster2 are not identified as Trailblazers, but the distinct gene expression profiles hinted at further examination. Specifically, we find that neural crest cells isolated shortly after dorsal neural tube exit (Cluster1) were enriched for genes known to be associated with early neural crest inductive signals and the neural plate border gene regulatory module previously described (*Simões-Costa and Bronner, 2015*) (*Figure 7A* and *Figure 4—source data 1*). These genes include *WNT3, WNT3A, WNT7B, WNT8B, ZIC1, ZIC3 MSX1, CDH6, EphA7*, and *CDH2*, some of which are highlighted in a volcano plot (*Figure 7A*).

In contrast, other known neural crest border plate genes (e.g., *TFAP2B, TFAP2C, TFAP2E* and *DLX5*) and neural crest specification and migration genes (e.g., *SNAI1, TWIST1, SOX10, ETS1, RXRG and EBF1, 2* and *3*) were reduced in Cluster1 (*Figure 7A* and *Figure 4—source data 1*). The transcription factor PAX6, which was not previously identified in the bulk RNA-seq analysis and neural crest gene regulatory network (*Simões-Costa and Bronner, 2015*) was enriched in HHSt11 Cluster1 cells, and is expressed in 85% of Cluster1 cells (*Figure 7A*). The HHSt11 Cluster2 subpopulation displays increased expression of genes related to later neural crest cell specification, epithelial-to-mesenchymal transition (EMT) and migration (*Figure 7B*). These genes include *PAX3, TFAP2B, SOX8* and *SOX10* (*Figure 7B* and *Figure 4—source data 1*). We also uncover additional transcription factors *SOX8* (expressed in all Cluster2 cells) and *FHL2* (expressed in 70% of Cluster2 cells) that were enriched in the HHSt11 Cluster2 subpopulation (*Figure 7B*). Furthermore, HHSt11 Cluster2 cells also showed decreased expression of factors thought to describe the neural crest gene regulatory network (e.g., *WNT2B, WNT3A, WNT4, WNT5B, WNT7B, WNT9A, BMPR1B, BMP2, BMP10, FGF3, FGF8, FGF10, ZIC1, TFAP2C, DLX1, DLX3, DLX6, SNAI1, ID1* and *ID3*) (*Figure 7B* and *Figure 4—source data 1*). A gene ontology (GO) analysis was compiled using the R package cluster Profiler (3.2.6). GO term analysis reveals distinct molecular characteristics of cells within each cluster (*Figure 7C–D*). Neural crest cells within Cluster1 display genes enriched in signaling, adhesion and cell guidance, with no depleted GO terms (*Figure 7C*). Cluster2 identifies with neural crest cells that display genes depleted in signaling and transcription regulation, with no enriched GO terms

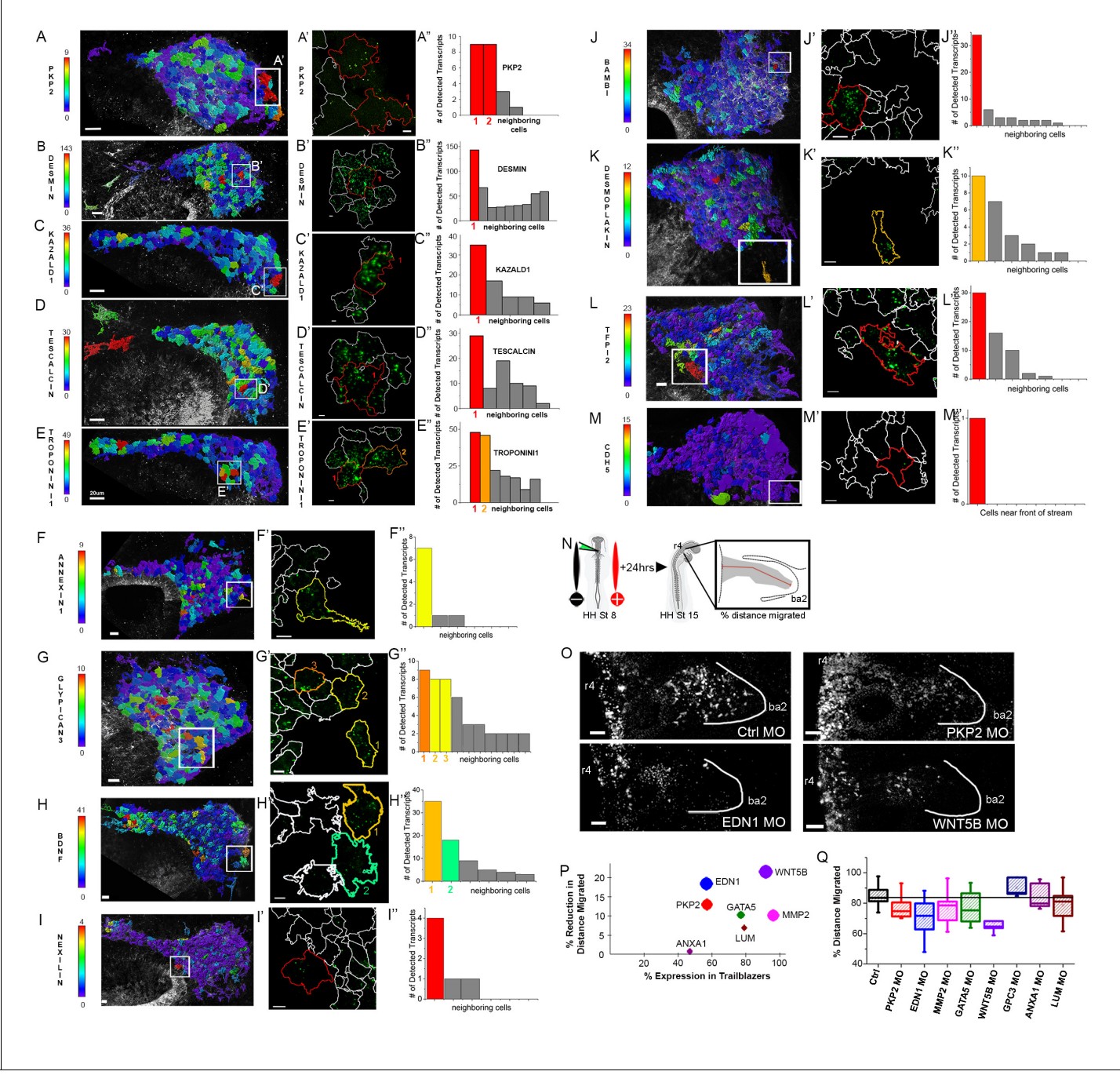

**Figure 6.** Expression and functional perturbations of selected genes enriched in Trailblazers. (A–M) Typical cranial neural crest cell migratory streams at HHSt13 adjacent to rhombomere 4 (r4) with HNK-1-positive cells color coded by the number of PKP2 (Plakophilin 2; A), Desmin (B), KAZALD1 (Kazal Type Serine Peptidase Inhibitor Domain1; C), Tescalcin (D), Troponin I1 (E), Annexin1 (F), Glypican 3 (G), BDNF (Brain Derived Neurotrophic Factor; H), Nexilin (I), BAMBI (BMP and Activin Membrane Bound Inhibitor; J), Desmoplakin (K), TFPI2 (Tissue Factor Pathway Inhibitor 2; L), and CDH5 (Cadherin 5; M). RNAscope spots detected per cell. (A'–M') Single cells outlined with expression were selected from each neural crest cell migratory stream and (A''–M'') the number of RNAscope spots per cell volume for a cell with high expression and adjacent neighboring cells is shown in the bar graphs. (N) Schematic representation of morpholino and electroporation procedure. (O) Morpholino transfected migrating cranial neural crest migration at HHST15 (n = 16 control MO embryos, n = 20 *EDN1* MO embryos, n = 20 *PKP2* MO embryos, n = 20 *MMP2* MO embryos, n = 7 GATA5 MO embryos, n = 5 WNT5B MO embryos, n = 10 GPC3 MO embryos, n = 7 ANXA1 MO embryos, n = 22 LUM MO embryos). (P) Correlation plot of distance migrated and expression in trailblazers. Circle size correlates with statistical significance: PKP2 MO, p=0.017, EDN1 MO, p=0.0004, MMP2 MO, p=0.0032, GATA5 MO, p=0.022, WNT5B MO, p=0.0001 (Q) Box plot of the distance migrated of morpholino transfected neural crest cells as a percentage of the distance

*Figure 6 continued on next page*

*Figure 6 continued*

from the neural tube to the tip of the branchial arch. HH, *Hamburger and Hamilton (1951)*; St, stage; OV, otic vesicle; MO, morpholino. Bar = 20 um (**A–G**). Bar = 15 um (**H–M**). Bar = 50 um (**O**).

DOI: https://doi.org/10.7554/eLife.28415.016

The following source data and figure supplements are available for figure 6:

**Source data 1.** Primer sequences used to test activity of splice blocking morpholinos.
DOI: https://doi.org/10.7554/eLife.28415.020
**Figure supplement 1.** Trailblazer genes chosen for perturbation experiments.
DOI: https://doi.org/10.7554/eLife.28415.017
**Figure supplement 2.** Testing of splice blocking morpholinos.
DOI: https://doi.org/10.7554/eLife.28415.018
**Figure supplement 3.** Analysis of stream neural crest migration after morpholino transfections in a subset of neural crest.
DOI: https://doi.org/10.7554/eLife.28415.019

(*Figure 7D*). These results more comprehensively describe the molecular characteristics of neural crest cells shortly after dorsal neural tube exit and suggest a dynamic transition from neural tube exit to acquisition of directed migration.

To identify signaling pathways enriched within HHSt11 clusters, we performed pathway enrichment analyses for the lists of genes differentially expressed by HHSt11 Cluster1 and Cluster2 versus all other in vivo NC cells to (*Figure 7E,F*). For both HHSt11 clusters, we found a mix of up- and down-regulated genes within these pathways, but overall observed increased gene expression associated with signaling pathways in Cluster1 and decreased gene expression associated with signaling pathways in Cluster2 (*Figure 7E,F*). Specifically, in Cluster1, 59% of differentially expressed genes associated with signaling pathways were up-regulated compared to only 1% in Cluster2. Interestingly, based on the gene expression, pathways such as WNT/Beta Catenin signaling, EMT, PCP pathway and axonal guidance signaling are increased in Cluster1 but then decreased in Cluster2 (*Figure 7E,F*). When focusing just on axonal guidance signaling, ADAMs, EPHs and WNTs are increased in Cluster1 but then decreased in Cluster2. The presence and absence of signaling pathways are also relevant. For example, Eph/ephrin signaling is significant in Cluster1 but not in Cluster2 (and therefore does not make the Cluster2 list) (*Figure 7E,F*). Actin cytoskeleton signaling is significant in Cluster2, with 21 out of 22 genes showing decreased expression, but this pathway is not significant in Cluster1 (*Figure 7E,F*). These results highlight the dramatic differences between Cluster1 and 2 at HHSt11 and suggest that their roles in EMT and migration may be very different.

## The transcriptional signatures of lead and trail subpopulations are defined by small numbers of differentially expressed genes

To define the genes and pathways associated with Lead and Trail subpopulations, we analyzed expression differences unique to each Lead and Trail subpopulation relative to all other in vivo cells isolated. Lead and Trail subpopulations were the largest subpopulations isolated (25% (Lead) and 70% (Trail) of the migratory stream,

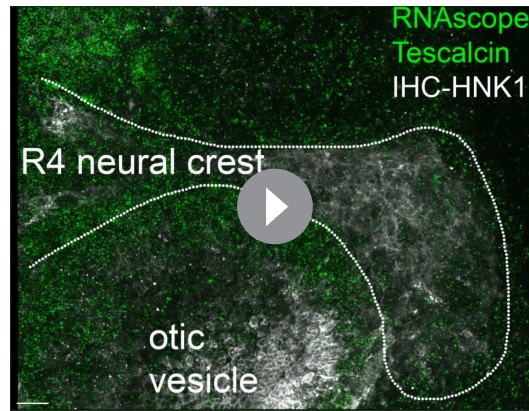

**Video 1.** RNAscope fluorescence signal of Tescalcin expression A typical 3D confocal image z-stack of a HHSt13 chick embryo highlighting the neural crest cell migratory stream adjacent to rhombomere four with cells labeled for Tescalcin by RNAscope and neural crest cells marked by HNK1 immunohistochemistry. The HNK1 membrane label was used to mark the boundaries of neural crest cells and mask the Tescalcin channel in 3D. The locations of RNAscope label for Tescalcin were detected and assigned to individual cells using the HNK1 boundaries. The cells were then colored by the number of detected Tescalcin RNAscope label spots with red being the highest and blue the lowest. A typical cell is approximately 20–30 um in diameter.

DOI: https://doi.org/10.7554/eLife.28415.021

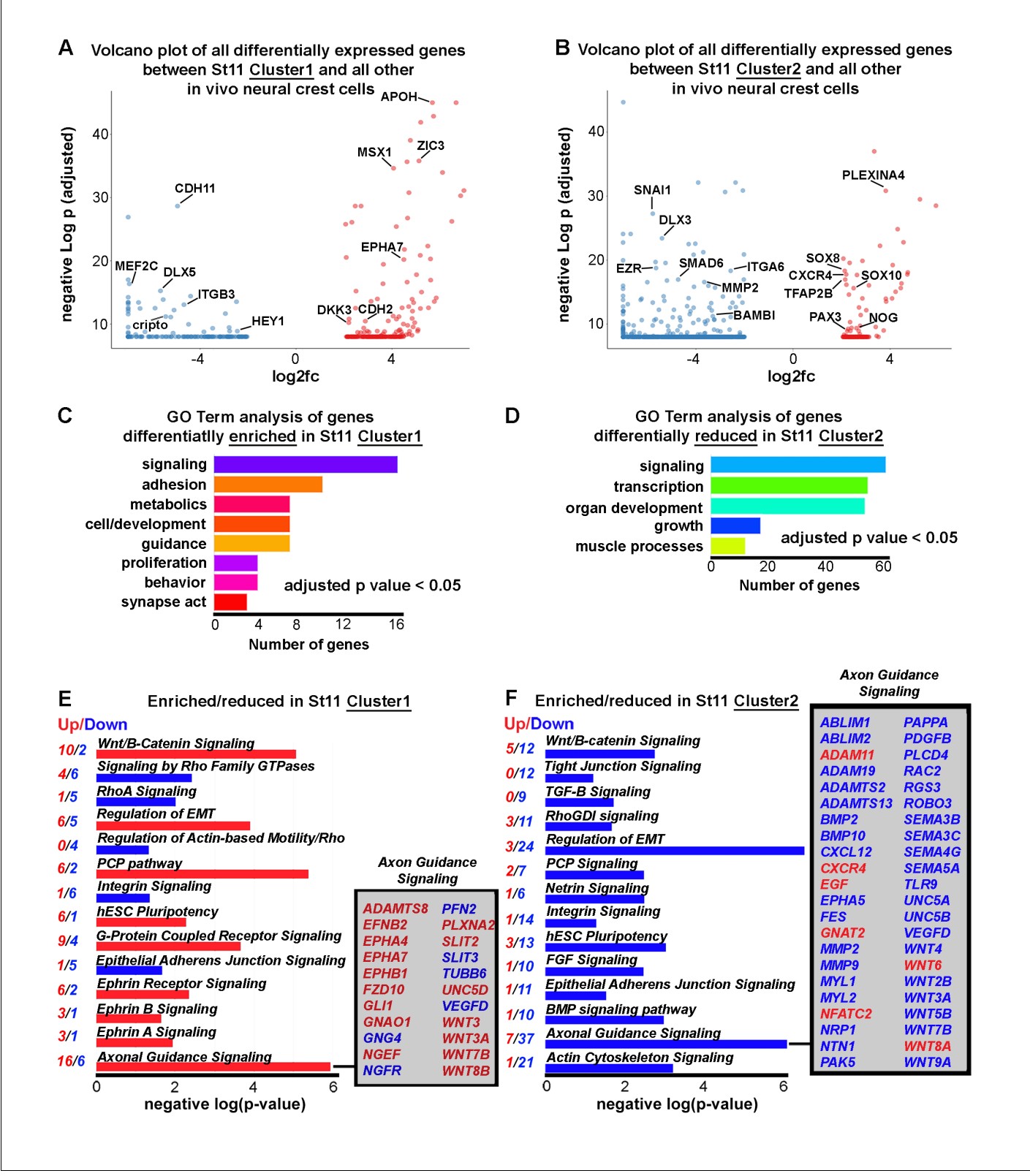

**Figure 7.** Newly emerging neural crest segregate into discrete groups representing a transition from EMT to directed migration. (**A**) Volcano plot of all genes statistically enriched or reduced in St11 Cluster1 (n = 20 cells, 381 genes). (**B**) Volcano plot of all genes statistically enriched or reduced in St11 Cluster2 (n = 55 cells, 1040 genes). (**C**) GO Term analysis of genes differentially enriched by HHSt11 cluster1. No GO terms were reduced. (**D**) GO Term analysis of genes differentially reduced by HHSt11 cluster2. No GO terms were enriched. (**E**) Significant signaling pathways predicted to be enriched

*Figure 7 continued on next page*

*Figure 7 continued*

(red) or reduced (blue) in St11 Cluster1. Axonal guidance signaling is highlighted. (**F**) Significant signaling pathways predicted to be enriched (red) or reduced (blue) in St11 Cluster2. Axonal guidance signaling is highlighted. GO terms with a p-value<0.05 and a q-value <0.1 were considered significant. Significant GO terms were considered using the simplify function with an adjusted p-value cutoff of 0.05.
DOI: https://doi.org/10.7554/eLife.28415.022

respectively), and exhibit high degrees of heterogeneity (*Figure 3* and *Figure 3—figure supplement 1*; *Figure 8—source data 1*). First, at HHSt13, we identified 45 enriched and 151 reduced genes in Trailers and 26 enriched and 35 reduced genes in Leaders (*Figure 8A,B*; *Figure 8—source data 1*). *AQP5*, *ALOX5AP*, and *SS5R* were enriched and *NKX2-6*, *NRG1* and *UNC5B* were reduced in Trailers (*Figure 8A*). *PKP1*, *PKDREJ*, and *CWH43* were enriched and *MSX1* and myosin reduced in Leaders (*Figure 8A*; *Figure 8—source data 1*). Strikingly, signaling pathway enriched analysis revealed reduced expression of genes associated with known developmental pathways (*Figure 5—source data 1*). These included VEGF signaling as well as genes associated with actin cytoskeleton and integrin signaling (*Figure 8C*).

Second, at HHSt15, we identified 174 enriched and 132 reduced genes in Trailers (*Figure 8D*; *Figure 8—source data 1*). Enriched genes in HHSt15 Trailers included *GATA5*, *HOXA3* and *NOG*; reduced genes included *FGF16*, *NEFM* and *NEUROD4* (*Figure 8D*; *Figure 8—source data 1*). Genes associated with developmental pathways such as Rho GTPases, RHOA, CXCR4, BMP and Actin cytoskeleton signaling were reduced in HHSt15 Trailers (*Figure 8F*; *Figure 5—source data 1*). HHSt15 Leaders had 77 enriched genes, including FGFs, *NKX2-6* and *SHH*, as well as 138 reduced genes, including *CDH7*, *PAX7* and *SOX10* (*Figure 8E*; *Figure 8—source data 1*). From these gene lists, signaling pathway enrichment analysis showed reduced activity in axonal guidance, ephrin receptor and netrin signaling pathways (*Figure 8F*; *Figure 5—source data 1*).

## Discussion

Cell migration in the embryo positions cells into precise target tissues at appropriate axial levels to ensure proper patterning and organ development. Directed motion and collective movement in groups are characteristics of embryonic cell migration and differ from processes where precursor cells are born and in the same location undergo shape changes and sorting to give rise to tissue structures. To study cell migration during embryonic development, we used the highly invasive neural crest as a model. We focused on cranial neural crest cell migration and exploited the strengths of the avian embryo to perform manual cell isolation at progressive developmental stages, bulk and single-cell RNA-seq to determine gene expression profiles within cells of a typical migratory stream over time. Our goal was to identify novel transcriptional signatures and molecular transitions associated with subpopulations of cells within a cranial neural crest cell migratory stream and determine how these transcriptional signatures changed as cells traveled through different microenvironments. This information has the potential to lead us to a better understanding of the molecular features underlying observed complex neural crest cell behaviors and mechanistically help to explain directed and collective cell migration processes.

We found that four features contribute to the precise directed and collective migration of the cranial neural crest to precise head targets. First, the majority of neural crest cells within the invasive front change their gene expression in a consistent manner during migration; however, a small subpopulation of Trailblazer cells narrowly confined to the invasive front have a conserved transcriptional signature throughout migration. Second, cranial neural crest cells newly exited from the dorsal neural tube cluster into two subpopulations with profiles distinct from the Trailblazer signature and indicative of a rapid switch from epithelial-to-mesenchymal transition to directed migration. Third, Lead neural crest cells (near the front of the stream, but not narrowly confined to the invasive front) cluster more closely with cells within the invasive front that do not display a Trailblazer transcriptional signature. Fourth, Trailing neural crest cells display a gene expression profile distinct from Lead and Trailblazer cells and cluster more closely with newly exited cells and in vitro migrating cells, the latter of which have a molecular signature distinct from any other in vivo profile.

The significant molecular distinction between cranial neural crest cells isolated from the invasive front and compared to other cells throughout the stream during migration (HHSt13 and HHSt15)

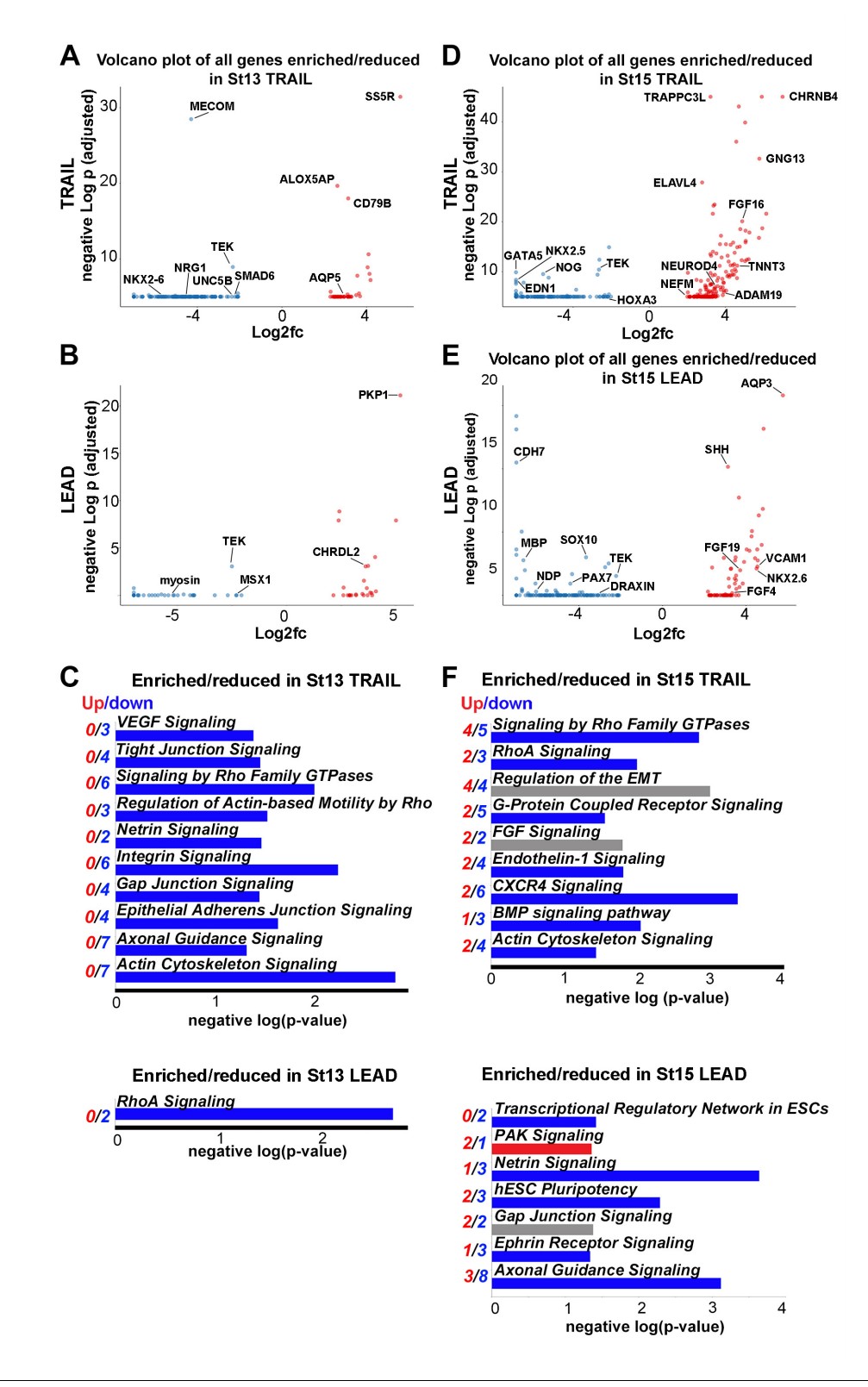

**Figure 8.** Lead and trailing subpopulations exhibit high degrees of heterogeneity. (**A**) Volcano plot of all genes statistically enriched or reduced in St13 Trail (n = 60 cells, 196 genes). (**B**) Volcano plot of all genes statistically enriched or reduced in St13 Lead (n = 49 cells, 61 genes). (**C**) Significant signaling pathways predicted to be enriched (red) or reduced (blue) in St13 Trail and Lead. (**D**) Volcano plot of all genes statistically enriched or reduced in St15

*Figure 8 continued on next page*

*Figure 8 continued*

Trail (47 cells, n = 306 genes). (**E**) Volcano plot of all genes statistically enriched or reduced in St15 Lead (n = 46 cells, 215 genes). (**F**) Significant signaling pathways predicted to be enriched (red) or reduced (blue) in St15 Trail and Lead.

DOI: https://doi.org/10.7554/eLife.28415.023

The following source data is available for figure 8:

**Source data 1.** Single-cell RNA-seq differential expression gene lists for HH13 and 15 Leader and Trailer subpopulations.

DOI: https://doi.org/10.7554/eLife.28415.024

suggests cells at the invasive front rapidly adapt to changes in the local microenvironments through which cells travel. Heatmaps of both the bulk RNA-seq 3X replicates (*Figure 1B*) and the mean values (*Figure 2—figure supplement 2*) showed that there are distinct clusters of genes expressed at significantly higher or lower levels in the invasive front when compared to cells within the remainder of the stream. The number of genes up-regulated in cells within the invasive front at HHSt13 (n = 61) and HHSt15 (m = 55) was consistent with an overlap that included 23 genes (*Figure 1*). In contrast, greater than 50% (255 out of 477) of the differentially expressed genes on the heatmaps (*Figure 1B*) at HHSt13 but less than 8% (36 out of 477) at HHSt15 were down-regulated in cells within the invasive front, with an overlap of 19 genes (*Figure 1B*, blue bands). The nearly 4-fold difference in the number of genes down-regulated versus up-regulated in cells within the invasive front at HHSt13 confirmed rapid changes take place as neural crest cells encounter the microenvironment of the branchial arch entrance. Yet, the 42 genes (23 up-regulated and 19 down-regulated) out of the 477 differentially expressed genes represented a conserved molecular signature at the two developmental stages and level of consistent gene expression during this microenvironmental transition.

In vivo expression analysis using RNAscope and immunolabeling identified heterogeneous expression within neighboring Trailblazers, but could not confirm gene expression unique to cells within the invasive front suggesting an inconsistency between the Trailblazer transcriptional signature and the in vivo mRNA expression patterns. While we expected the RNAscope expression analysis to identify more Trailblazer cells than were detected we found that none of the 13 genes tested were expressed in all of, or the majority of the cells at the invasive front. Instead, RNAscope analysis showed some genes either appeared in a few Trailblazer cells or displayed salt and pepper patterns that included strong expression in a few cells in the middle or back portion of the migratory stream (*Figure 6*). Further, some genes were not found expressed at the front edge at all (*Figure 6*). This could be due to differences in sensitivity between scRNAseq and RNAscope technologies. That is, the chemistry of scRNAseq may detect more transcripts than the newly established RNAscope recently optimized for avian embryos (*Morrison et al., 2017*). Further, the low number of cells within the invasive front that we detected to have a Trailblazer signature (for example, at HHSt13 there are approximately 10–15 Trailblazer cells out of 500 total cells in the migratory stream) make it difficult to visualize such a low number of expressing cells. In addition, the 3D nature of the cranial neural crest cell migratory stream and depth into the embryo make it difficult to more accurately assess fluorescence transcript levels in trailblazer cells in comparison to follower cells located more dorsal and accessible to signal detection by confocal microscopy. Future studies aided by technological advances that increase resolution may better link scRNA-seq data with mRNA expression detection.

Several genes discovered within the scRNA-seq Trailblazer transcriptional signature and previously identified in our RT-qPCR analysis have been implicated in human cancer cell invasion. These genes included *AQP1*, *BAMBI*, *CTTNB1*, and *PKP2* (*Figure 5*; *McLennan et al., 2015*). *BAMBI*, a transforming growth factor beta (*TGFB*) inhibitor is up-regulated in metastatic carcinomas including colorectal and osteosarcoma (*Fritzmann et al., 2009*; *Zhou et al., 2013*). Similarly, the expression of *PKP2*, a desmosomal adhesions protein has been implicated in glioma and bladder cancer metastasis (*Takahashi et al., 2012*; *Zhang et al., 2017*) and high expression of Tescalcin, a gene thought to play a role in cell differentiation and implicated in invasive colorectal cancer (*Kang et al., 2016*). Furthermore, *WNT5B*, a gene identified within our scRNA-seq Trailblazers has been implicated in cancer aggressiveness in a number of cancer cell lines and tissues (*Harada et al., 2017*). When we knocked down *PKP2* and *WNT5B* in premigratory neural crest cells by morpholino we found significant reduction in cell distance migrated (*Figure 6N–Q*; *WNT5B* (21% reduction, p=0.0001), *PKP2* (13% reduction, p=0.017). Thus, genes identified and associated with some of the most invasive

cranial neural crest cells may have a critical functional role in migration and may be part of a broader molecular signature in cancer aggressiveness.

Pathway analysis of the Trailblazer transcriptional signature identified genes involved in VEGF signaling, strengthening support for our previous discovery that VEGF chemoattraction is critical for neural crest migration (*McLennan and Kulesa, 2007*; *McLennan and Kulesa, 2010*; *McLennan et al., 2010*). We found that all three VEGF receptors as well as neuropilin-1 (*NRP1*) were expressed at significantly higher levels in the Trailblazer neural crest cells (*Figure 5D*). When we previously knocked down neuropilin-1 in chick cranial neural crest cells, cells were able to migrate away from the neural tube but failed to invade the second and rostral portion of the third branchial arches (*McLennan and Kulesa, 2007*; *McLennan and Kulesa, 2010*; *McLennan et al., 2010*). These data provide strong evidence to support a critical role of the Trailblazer neural crest cells to guide invasion into the branchial arches.

Axonal guidance signaling in general has provided insights that have led to discoveries of mechanisms of neural crest migration in the head (*Osborne et al., 2005*; *Gammill et al., 2007*), eye (*Lwigale and Bronner-Fraser, 2009*) and gut (*Anderson et al., 2007*). Furthermore, there is exciting emerging evidence that genes implicated in nervous system development have functional similarities with vascular patterning, including endothelial tip cell invasion (*Wälchli et al., 2015*). When we performed pathway analysis, we found a significant number of axonal growth cone signaling genes (n = 26) identified within the Trailblazer transcriptional signature (*Figure 5D*) and early emigrating neural crest cells (*Figure 8*) indicative of shared molecular features between the two invasion phenomena. However, our investigation of the neural crest literature showed that only 7 out of the 26 genes we found enriched in the Trailblazer cells have been examined in neural crest cell migration, including *BMP2, BMP4, MMP2, RAC2, SEMA3C, WNT1*, and *WNT11* (*Anderson, 2010*; *Correia et al., 2007*; *de Melker et al., 2004*; *Goldstein et al., 2005*; *Matthews et al., 2008*; *Tang et al., 2016*; *Toyofuku et al., 2008*). Thus, determination of the function of the other 19 out of 26 axonal growth signaling genes in neural crest migration may lead to exciting insights and further our knowledge of common developmental principles of neurovascular patterning (*Figure 5D*).

The scRNA-seq analysis validated some but not all of the genes identified by our 16 gene RT-qPCR Trailblazer signature (*McLennan et al., 2015*), which is understandable given the key variances between the two analyses. Specifically, the RT-qPCR Trailblazer signature was defined by cells narrowly confined within the invasive front in HHSt13 and 15 embryos. Our scRNA-seq data defined Trailblazers as a molecularly distinct subpopulation within the invasive front at HHSt11, 13 and 15. Second, RT-qPCR and scRNA-seq are distinct methodologies. Notably, RT-qPCR is the gold standard of sensitivity for detecting lowly abundant transcripts, whereas scRNA-seq is capable of uncovering many more genes within a single analysis. Thus, our new scRNA-seq analysis added information from HHSt11 cells to refine and expand the Trailblazer signature based upon unbiased clustering of a comprehensive transcriptome analysis.

Newly emerging neural crest cells clustered as two distinct subpopulations with transcriptional signatures unique from that of the Trailblazer cells suggesting a rapid molecular transition to directed migration shortly after neural tube exit. At HHSt11, there are few neural crest cells that have delaminated from the dorsal neural tube at the axial level of r4 such that we grouped together all cells that were isolated (*Figure 2*). Hierarchical clustering identified the Trailblazer signature and two other transcriptional signatures that we associated with Cluster1 and Cluster2 (*Figure 4*). Genes within the transcriptional signature of Cluster1 closely resembled those related to the premigratory and delamination events of the neural crest (*Figure 7*). In contrast, genes within the transcriptional signature of Cluster2 more closely resembled those related to directed neural crest cell migration (*Figure 7*). Interestingly, we found very little overlap between the Cluster1 and Cluster2 transcriptional signatures that was indicative of a rapid switch in the gene expression profile rather than continuous progression of gene expression changes (*Figure 7*). Further analysis of mRNA and protein expression of multiple genes using RNAscope as shown above or in other emerging strategies (*Choi et al., 2016*; *Roellig et al., 2017*) will help to shed light on the spatial positions of cells with these unique transcriptional signatures identified by scRNA-seq.

The significant differences in gene expression profiles comparing in vivo transcriptional signatures with in vitro data was not surprising given the absence of the neural crest microenvironment and lack of formation of discrete migratory streams in culture. PCA analysis highlighted the poor correlation in gene expression patterns (*Figure 2C,D*) with the closest subpopulation to the in vitro neural

crest cells being a subpopulation of the newly exited neural crest cells at HHSt11 (*Figure 2C*). This made logical sense since both in vitro and newly exited in vivo neural crest cells are within the range of influence from signals within the neural tube (*Figure 2C*). Furthermore, no GO terms were upregulated in in vitro neural crest cells and instead there was a down-regulation of terms such as development, cell signaling, migration and proliferation (*Figure 2E*). Furthermore, our previous RT-qPCR study showed that migrating chick cranial neural crest cells did not display distinct Lead and Trailer gene expression profiles in culture (*McLennan et al., 2015*). However, exposure to VEGF in vitro resulted in the upregulation of a small subset of genes associated with an in vivo lead cell signature (*McLennan et al., 2015*). Together, this evidence supported our model that the in vivo embryonic neural crest cell microenvironment critically influences transcriptional signatures of migrating neural crest cells.

Advances in cell isolation and single-cell RNA-seq have proven invaluable for elucidating cell-type specific transcriptional signatures, often defined by an ensemble of genes rather than high expression of any single gene (*Dueck et al., 2016*; *Tasic et al., 2016*), during single time-point embryonic and adult neurogenic events (*Chu et al., 2016*; *DeLaughter et al., 2016*; *Nelson et al., 2016*; *Scholz et al., 2016*; *Tirosh et al., 2016a*, *2016b*; *Dulken et al., 2017*; *Litzenburger et al., 2017*). However, the identification of distinct transcriptional signatures associated with dynamic cell populations during cell migratory events has remained challenging and unclear. By using single-cell transcriptome analysis applied to embryonic cranial neural crest cell migration at three progressive developmental stages, we provide the first comprehensive analysis of neural crest migration. Our data identify and establish hierarchical relationships between cell position- and time-specific transcriptional profiles, including a Trailblazer transcriptional signature characterized by a large number of differentially expressed genes within a small subpopulation of cells at the invasive front. Functional in vivo knockdown of a subset of individual Trailblazer genes showed significant but modest changes in neural crest cell distance migrated. However, expression analysis by RNAscope multiplex fluorescence in situ hybridization could not definitively confirm expression restriction to the invasive front. The present study provides new insights into the molecular patterns that underlie directed, collective cell migration of the neural crest and reveals individual targets and signaling pathways for future mechanistic studies and to other neural crest migratory streams throughout the body and other cell invasion phenomena in cancer, wound healing, and the immune system response.

## Materials and methods

### In Ovo electroporation

Fertilized, white leghorn chicken eggs (NCBI Taxonomy ID:9031; Centurion Poultry Inc., Lexington, GA, USA) were incubated in a humidified chamber at 38C to the developmental desired stage. For NC cell isolations, pre-migratory NC were labeled by electroporation of a Gap43-YFP plasmid DNA into the dorsal neural tube of HHSt9 embryos as previously described (*McLennan and Kulesa, 2007*) and re-incubated (8, 16 and 24 hr for HHSt11, 13 and 15, respectively). After re-incubation, embryos were harvested into chilled 0.1% DEPC PBS and screened for health and transfection efficiency before tissue isolations were performed.

For functional analysis, H2B mCherry (2.5 µg/µl) and fluorescein-tagged morpholinos (0.5 mM) (Gene Tools, Philomath, OR, USA) were injected into the neural tube and electroporated at HHSt8 (*Figure 6N*). The embryos were re-incubated to HHSt15. After re-incubation, embryos were harvested and fixed in 4% paraformaldehyde at room temperature for 2 hr or at 4C overnight. Immunohistochemistry was then performed on the embryos for HNK1 (neural crest cell marker) using a standard protocol (*McLennan and Kulesa, 2007*) before being imaged on a LSM 800, using a Plan-Apochromat 10x/0.45 M27 objective (Zeiss, Oberkochen, Germany) (*Figure 6—figure supplement 3*).

For splice-blocking morpholino validation by RT-PCR, the chicken LMH cell line was transfected with either control or experimental splice blocking morpholino using Endo-porter (Gene Tools, Philomath, OR). 48 hr after transfection, cell lysate was harvested and total RNA isolated (RTN70 Sigma, St. Louis, MO). RNA was converted to cDNA (4368814, Thermo Fisher Scientific, Waltham, MA) for RT-PCR (M7122, Promega, Madison, WI, USA) using primers flanking the region affected by each morpholino to demonstrate perturbation of translation (*Figure 6—source data 1*). PCR products

were analyzed on a LabChip GX Touch (PerkinElmer, Waltham, MA) (*Figure 6—figure supplement 2*).

## Tissue isolations

All manual dissections of Gap43-YFP transfected neural crest cells were performed using a fluorescent dissecting microscopy. For bulk RNA-seq samples, the NC streams adjacent to rhombomere 4 (r4) were isolated by manual dissection from HHSt13 and 15 embryos and divided into three biological replicates. The streams were subsequently cut into two spatially distinct, non-overlapping portions: the most invasive front (5% of the stream, termed 'Front') and the remaining proximal-most trailing portion of the stream (95% of the stream, termed 'Stream'). Developmental stage- and stream position-matched pools of tissue were dissociated as previously described (*Morrison et al., 2015*).

For scRNA-seq, the NC r4 stream was isolated by manual dissection at HHSt11, 13 and 15. HHSt11 r4 streams containing few migrating NC were pooled. For HHSt13 and 15 embryos, the r4 NC stream was further subdivided by manual dissection into three spatially distinct, non-overlapping portions: the most invasive front (5% of the stream, termed 'Front'), the leading portion of the stream (25% of stream proximal of the most invasive front, termed 'Lead') and the proximal-most trailing portion of the stream (70% of the stream, termed 'Trail'). Developmental stage- and stream position-matched tissues were pooled and dissociated before cytometric isolation as previously described (*Morrison et al., 2015*). Optimizing tissue dissociations and limiting processing time from embryo to lysis are critical for maintaining cell health and generating high quality scRNAseq data (*Morrison et al., 2015*).

## In Vitro neural crest cell cultures

Cranial neural tubes (NT; r3-r5) were removed from embryos at HHSt9, plated as previously described (*Krull et al., 1997*). NT were incubated for 24 hr to allow NC to emerge from the NT and migrate onto the substrate. NT were then gently removed with a tungsten needle, so that only migratory NC remained. Media was gently removed and the adherent cells rinsed once with 0.1% DEPC PBS before trypsinization (25200–056, Thermo Fisher Scientific, Waltham, MA) for 5 min at 37C and trypsin inactivation with fetal bovine serum. The cells were then pelleted and resuspended in chilled 0.1% DEPC PBS.

## Cytometry

Cells were isolated by FACS, which included forward scatter, side scatter, pulse width, live/dead stain and YFP gates as previously described (*Morrison et al., 2015*). Cells were sorted directly into 6 ul of Clontech lysis solution containing 0.05% RNAse inhibitor. Following lysis for 5 min at room temperature, lysates were immediately frozen on dry ice and stored at −80C.

## Bulk RNA-seq

Bulk RNA-seq lysates were thawed on ice. cDNA synthesis and library preparation were performed with SMART-seq v4 Ultra Low Input RNA-seq (634892, Takara, Kusatsu, Shiga, Japan) and Nextera XT DNA dual indexing library preparation kits as recommended by the manufacturer (FC-131–1096, Illumina, San Diego, CA). All 12 libraries (4 samples with 3 biological replicates each) were pooled and run across 2 lanes of an HiSeq 2500 generating 50 bp single reads (Illumina). Quality control analyses, including reads per samples, coverage by normalized position along transcript and correlation among all samples and biological replicates are shown in *Figure 1—figure supplement 1*.

## Single-cell RNA-seq

Single-cell lysates were thawed on ice. cDNA synthesis and library preparation were performed with SMART-seq v4 Ultra Low Input RNA-seq (634892, Takara, Kusatsu, Shiga, Japan) and Nextera XT DNA dual indexing library preparation kits as recommended by the manufacturer (FC-131–1096, Illumina). Despite more cost-effective scRNAseq strategies, SMARTseq v4 was employed in order to detect the greatest number of genes per single cell, which is critical for distinguishing highly similar cell subpopulations (*Ziegenhain et al., 2017*; *Dueck et al., 2016*). All single-cell cDNAs and libraries were first quantified on a Qubit. Representative single-cell cDNAs and libraries were also quantified

on a Bioanalyzer 2100 High Sensitivity DNA Assay (Agilent, Santa Clara, CA). A maximum of 96 single-cell libraries were pooled per lane of a HiSeq 2500 generating 50 bp single reads (Illumina). Quality control analyses for scRNA-seq are shown in *Figure 2—figure supplement 1*.

## Computational analysis

Fastq files were mapped to the chicken genome galGal4 from UCSC using Tophat (2.0.13) with options -x 1 g 1. Ensembl 80 annotations were used to define gene coordinates. Quality of the samples was assessed using FastQC (0.10.1). The R environment (RRID:SCR_001905) was used for the statistical analysis of the data. All annotated code is located in *Supplementary file 1*. RPKM values generated and differentially expressed genes were found using edgeR. Genes were considered differentially expressed if they had an Benjamini-Hochberg adjusted p-value less than 0.05 and a log2 fold change greater than 2 or less than −2. Hierarchical clustering was performed using Pearson correlations. Clustering was achieved using base principal component functions in R. Gene ontology (GO) analysis was completed using ClusterProfiler (3.2.6). GO terms with a p value less than 0.05 and a q value below 0.1 were considered significant. Significant GO terms were condensed using the simplify function with an adjusted p value cutoff of 0.05. Pathway enrichment analyses were performed using Qiagen's Ingenuity Pathway Analysis (RRID:SCR_008653; v31813283, Hilden, Germany).

## RNAscope ISH, Immunohistochemistry and Image Analysis

RNAscope multiplex fluorescent ISH (Advanced Cell Diagnostics, Newark, CA) combined with IHC in the whole avian embryo was adapted from Gross-Thebing et al.'s work in the zebrafish (*Morrison et al., 2017*; *Gross-Thebing et al., 2014*). Major optimizations for success in the avian included fixation time, avoiding air drying the embryos, volumes and incubation times. IHC to label migratory NC in the whole embryo was performed as described previously (*McLennan and Kulesa, 2007*). Images were collected on a LSM 800 confocal microscope. Images were analyzed with the Cell module in Imaris (RRID:SCR_007370; Bitplane, Belfast, Northern Ireland). Using the HNK1 as a neural crest membrane marker (RRID:AB_10013722), cells were segmented and then spots of RNAscope signal detected and assigned to each cell. Smoothing and background subtraction aided in segmentation of the cells. Manual modifications to the cell borders was completed in some cases where Imaris failed to segment properly using the Surface module and marching cubes detection. Cell surfaces were colored to represent the number of detected transcripts.

For morpholino transfected fixed embryos at HHSt15, a sharpened tungsten needle was used to remove the trunk portion of the embryo and cut the cranial embryo in half down the midline. Embryos were then mounted into glass slides as previously described (*Teddy and Kulesa, 2004*) and imaged by confocal microscopy on a LSM 800 (Zeiss). The percentage of distance the morpholino transfected cells migrated was calculated by measuring the total distance of the ba2 and distance the morpholino transfected cells migrated.

## Acknowledgements

This research was supported by the generosity of the Stowers Institute for Medical Research and partially supported by the National Institute of Neurological Disorders and Stroke as well as the National Institute of Child Health and Human Development of the National Institutes of Health under Award Numbers R21NS092001 and R03HD089190, respectively. We also thank members of the Flow Cytometry, Molecular Biology and Bioinformatics core facilities at the Stowers Institute. Original data underlying this manuscript can be accessed from the Stowers Original Data Repository at http://www.stowers.org/research/publications/LIBPB-1158.

## Additional information

### Funding

| Funder | Grant reference number | Author |
| --- | --- | --- |
| Stowers Institute for Medical Research | | Paul M Kulesa |

| National Institute of Neurolo-gical Disorders and Stroke | R21NS092001 | Paul M Kulesa |
| National Institute of Child Health and Human Develop-ment | R03HD089190 | Paul M Kulesa |

The funders had no role in data collection and interpretation, or the decision to submit the work for publication.

## Author contributions

Jason A Morrison, Conceptualization, Formal analysis, Supervision, Funding acquisition, Validation, Investigation, Visualization, Methodology, Writing—original draft, Project administration, Writing—review and editing; Rebecca McLennan, Conceptualization, Data curation, Formal analysis, Supervision, Validation, Investigation, Visualization, Methodology, Writing—original draft, Project administration, Writing—review and editing; Lauren A Wolfe, Conceptualization, Data curation, Software, Formal analysis, Supervision, Validation, Investigation, Visualization, Methodology, Writing—original draft, Project administration, Writing—review and editing; Madelaine M Gogol, Software, Formal analysis, Validation, Methodology, Writing—original draft; Samuel Meier, Software, Formal analysis, Validation, Investigation, Methodology; Mary C McKinney, Jessica M Teddy, Software, Formal analysis, Validation, Investigation, Visualization, Methodology; Laura Holmes, Craig L Semerad, Andrew C Box, Data curation, Software, Validation, Investigation, Methodology; Hua Li, Kathryn E Hall, Data curation, Software, Investigation, Visualization, Methodology; Anoja G Perera, Data curation, Supervision, Methodology; Paul M Kulesa, Conceptualization, Data curation, Supervision, Funding acquisition, Methodology, Writing—original draft, Project administration, Writing—review and editing

## Author ORCIDs

Jason A Morrison https://orcid.org/0000-0002-1654-8076
Madelaine M Gogol https://orcid.org/0000-0002-8738-0995
Paul M Kulesa https://orcid.org/0000-0001-6354-9904

## Ethics

Animal experimentation: All experiments were performed according to institutional (IBC-2003-23-pmk) and federal ethical standards.

## Decision letter and Author response

Decision letter https://doi.org/10.7554/eLife.28415.028
Author response https://doi.org/10.7554/eLife.28415.029

# Additional files

## Supplementary files

• Supplementary file 1. Annotated code for all bioinformatic analysis
DOI: https://doi.org/10.7554/eLife.28415.025

• Transparent reporting form
DOI: https://doi.org/10.7554/eLife.28415.026

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
