## [Decision Letter]

[Editors’ note: a previous version of this study was rejected after peer review, but the authors submitted for reconsideration. The first decision letter after peer review is shown below.]

Thank you for choosing to send your work, "Single-cell transcriptome analysis of neural crest migration reveals signatures of invasion and molecular transitions", for consideration at *eLife*. Your initial submission has been assessed by a Senior Editor, who is also a member of our Board of Reviewing Editors, and two reviewers. Although the work is of interest, we regret to inform you that the findings at this stage are too preliminary for further consideration at *eLife*.

Specifically, the reviewers think you should confirm expression of some of trailblazer genes in embryos and perform further in vivo tests for a few more candidate genes. In addition, the revised manuscript should compare the new data with your results previously published in 2015, to establish if single cell RNA-seq approach agrees with or changes our perspective. It would be important to emphasize what is new. We would be very interested in receiving a new version of this paper after addressing all of the reviewers' comments and every effort would be made to return the paper to the same reviewers. I have included the full reviews which I hope will be helpful in revising the manuscript.

*Reviewer #1:*

This study, titled "Single-cell transcriptome analysis of neural crest migration reveals signatures of invasion and molecular transitions" submitted by Jason Morrison et al., is a follow up on their group study in 2015 (McLennan et al., 2015 Development 142:2014-25), uses transcriptomic analyses to dissect the molecular signatures of the known and novel subpopulations within the avian neural crest migratory streams. The study identifies molecular signatures for the 'trailblazer', a subpopulation of cells at the invasive front of the migratory stream, and two distinct subpopulations of neural crest cells exiting the dorsal neural tube. The study partially confirms previous findings in terms of the trailblazer molecular signatures but also raises questions regarding the experimental designs and analyses that should be performed to engage in an unbiased manner the transcriptomic heterogeneity of the migratory neural crest cells (see comments below). The reviewer found a scarcity of functional assays to explore the biological meaning of cellular heterogeneity in neural crest (only one supplemental figure described in the Discussion section, which perhaps should be brought to the main portion of the manuscript). It seems fundamental to verify the RNA seq data sets by independent assays, e.g., PCR or in situ, on key genes/pathways that had been identified in the bioinformatics analyses, as well as with the inclusion of known markers for each relevant population. The reviewer also suggests removal of some figure panels that appear to be irrelevant to the study to streamline the flow of the manuscript (see comments below).

Results section 1

• Why were these two stages being chosen? Stage 13 appears to be very close to stage 15 in terms of gene expression. Bioinformatics analyses also did not appear to justify there was a stage specific difference.

• At what stage did the authors label the premigratory neural crest cells?

• There was no description whether the 16 genes identified in McLennan et al., 2015 were found in the bulk RNA seq and what were the new players. For example the analyses should have been performed on genes such as, Hand2/GPC3 as invasive front signature, PDGFRL/CXCR1 as stream signature, and EPHA4 and FOXD3 as trailing signature identified in McLennan et al. 2015. These need to be first established in the bulk RNA seq analysis.

Results section 2

• "we isolated and profiled individual cells from different stream positions at three developmental stages". “We subdivided neural crest cell migratory streams at HHSt13 and HHSt15 into three spatially distinct, non-overlapping subregions by careful manual dissection and single cells were isolated by FACS". The authors need to justify the rationales behind the selection of stages (two versus three) and dissection methods (front/trail vs. front/lead/trail) for the bulk and single cell RNA seq. Were there other previous experiments and studies that the authors or others had performed that would help justify the differences?

• "Hierarchical clustering of single neural crest cell averages….." The initial analysis should have been performed without averaging the single cell transcriptome values according to their migration positions. The current experimental design, by clustering single cell samples, could have unavoidably biased the results. The conclusion reached also did not add any new insight into the study. Figure 2 could be removed. The aim of the experiment was to identify new genes/signatures that can define the different subpopulations and the authors could have used the previous 16 genes (McLennan et al., 2015) or other known targets from the bulk RNA seq in Figure 1 or other studies as a guide to distinguish the front cells.

Results section 3

• "in vitro transcriptional signatures are distinct from any in vivo profile". This section appears not relevant to the rest of the study. This section does not give any new insight into the cellular heterogeneity of migrating neural crest cells.

Rest of the Results sections

• “Our previous RT-qPCR (McLennan et al., 2015) and bulk RNA-seq presented above (Figure 1) show that cranial neural crest cells at the invasive front of a stream have aspects of their transcriptome that are conserved during migration.” This rationale does not fit into the experimental design for Figure 5. If the invasive front were to have a conserved transcriptome during migration, then all cells should have been clustered for analysis. The experiment presented in Figure 5 instead appears to try to identify heterogeneity of the front cells dissected from the embryos.

• Figure 5: Where did the front cells for stage 11 come from? It was stated in the first paragraph of the subsection “Single-cell RNA-seq identifies gene expression variances based upon spatial position within the neural crest cell stream and temporal progression along the migratory pathway” that there was no subdivision of stage 11 cells.

• Use different colors for Trail and Trailblazers in Figure 6.

• Subsection “The scRNA-seq Trailblazer signature is uniquely defined and associated with neural crest cells confined to the invasive front of the stream”, last paragraph: It is interesting that the authors were able to identify previously known pathways, such as the VEGF signaling, that were associated with Trailblazer identity. However, as the purpose was to discover new pathways, an unbiased search should be performed instead of a focus on developmental relevant signaling pathways. As noted, a significantly higher number of pathways were found to be upregulated and many of the p-values were relatively high (Figure 6) suggesting that there might exist other "developmentally irrelevant" pathways that were associated with trailblazer signatures.

• Please highlight the p-value in Figure 6 that represent significance difference.

• Figure 6 can be better presented with the p-values align according to their values from top to bottom.

• What were the criteria to select the gene list for presentation in Figure 6? Were these genes most highly enriched or depleted?

• Highlight in the text and figure legend for Figure 6 that those genes highlighted in red were previously found in McLennan et al., 2015.

• The conclusion "These results highlight the upregulation of many genes within the trailblazer subpopulation, supporting the hypothesis that trailblazers are a dramatically active subpopulation of cells at the most invasive front during neural crest migration" needs to be substantiated (and if possible reworded to omit superlatives.

• " We also found that CDH7, CXCR4 and EPHB1 were upregulated in the HHSt11 single cells (Figure 6)." What were these single cells? Were there any bias in stage 11 clusters 1 and 2 identified earlier?

• Subsection “The scRNA-seq analysis validates some of our original 16 gene RT-qPCR Trailblazer signature”: the authors said there was "no significant enrichment" for a number of genes found in previous studies meaning the expression of these genes were upregulated but not statistically significant or were downregulated instead? Following up on this, if these genes were not upregulated in the trailblazer population, were they found to be upregulated in any of the subpopulations of the migrating neural crest cells?

• Subsection “Identification of two distinct transcriptional signatures associated with neural crest cells isolated shortly after dorsal neural tube exit”, second paragraph: How was the GO search performed for clusters 1 and 2? Why were there no depleted and enriched terms for clusters 1 and 2 respectively?

• Please correct typos. “To identify signaling pathways enriched within HHSt11 clusters, we performed pathway enrichment analyses "for" the lists of genes differentially expressed by HHSt11 Cluster1 and Cluster2 versus all other in vivo NC cells to[…]”.

• "There may be even more dramatic differences between the pathways associated with each cluster in the form of significance." It is not clear what was the message the authors were trying to convey in this sentence.

• Subsection “Identification of two distinct transcriptional signatures associated with neural crest cells isolated shortly after dorsal neural tube exit”, last paragraph: It is intriguing to find that WNT/PCP signaling were respectively up and downregulated in clusters 1 and 2. It might shed important functional roles of these distinct subpopulations of stage 11 neural crest cells by altering WNT at this stage and to see if and how skewed production of these subpopulations lead to phenotypes in neural crest cell migration and formation of the Trailblazer population at later stages.

• "This conclusion is most likely the result of increased heterogeneity of these subpopulations due to the increased size when compared to the other subpopulations examined." The opposite appears to be true to common sense: reduced sample size will likely lead to increased heterogeneity. The findings presented here suggest that trail and lead subpopulations were highly heterogeneous and no defined signatures can be identified. Increasing the sample size (number of cells) in future experiments on the other hand might help to identify small subpopulations.

• The result description for Figure 6—figure supplement 1 should be elaborated and should be placed within the Results section instead of in the DDiscussion section.

*Reviewer #2:*

In the manuscript "Single-cell transcriptome analysis of neural crest migration reveals signatures of invasion and molecular transitions", Morrison and colleagues employ transcriptional profiling of individual neural crest cells to investigate cell heterogeneity during migration. The authors begin by performing bulk RNA-seq experiments to identify transcripts enriched and depleted in the invasive front. Next, they survey the transcriptome of single neural crest cells to establish the molecular signatures of distinct subpopulations within the R4 stream (also showing that there are substantial molecular differences between cells collected directly form embryos or cultured in vitro). The study is centered on the characterization of the transcriptional signature of trailblazers (cells that lead the neural crest migratory stream) and the molecules that set this subpopulation apart.

I found the manuscript to be well structured, given the challenge of condensing and organizing transcriptomic data from hundreds of cells. This is partly due to the decision (a good one, in my view) of focusing specifically on the differences between leading vs. trailing cells. Indeed, the manuscript represents a substantial addition to the work the group has done on this topic – while also resulting in a useful resource for future studies. Nevertheless, I believe it would greatly benefit from additional in vivo analysis. While reading the paper, I found myself trying to visualize how the distinct populations characterized through transcriptomics were distributed within the R4 stream. Where would the Trailblazer cells be and how many of them would be present? Mapping the expression of the novel trailblazer genes back into the embryo would be an important validation of the analysis, and yield in spatial information that could clarify the molecular heterogeneity observed.

1) How do the authors reconcile the results from the bulk to the single cell experiments? I would expect to find the genes that are part of the Trailblazer molecular signature to be enriched in the invasive front, but that is only the case for a handful of genes. Is this a matter of sensitivity / sequencing depth?

2) The use of single-cell transcriptomics allowed the authors to refine their previous findings on the molecular signature of the trailblazer cells. Yet I found the discussion on the significance of this signature to be somewhat vague. Are Trailblazer cells characterized by the co-expression of these genes? Are there any Trailblazer-specific individual factors? How heterogeneous is the Trailblazer population? In short – if one needs to identify Trailblazer cells, what are the factors they should be focusing on?

3) While I appreciate the authors' use of microdissection to incorporate a degree of spatial resolution in the analysis, the work would greatly benefit from additional in vivo experiments. The authors should confirm co-localization of at least some of the transcripts that are part of the trailblazer molecular signature in embryonic tissue. Is it possible to map individual trailblazer cells utilizing these markers? How large is the population of trailblazers in the stages analyzed, and is this consistent with the results of the single cell transcriptome data?

4) The morpholino experiments on Figure 6 come across as an afterthought. Have these morpholinos been published (I do not see the morpholino sequences in the method sections). How are these genes expressed? Proper controls for efficiency and specificity should also be included.

5) Surprisingly, Trailblazing cells have weaker expression of genes that are thought to be central to the neural crest regulatory program such as Tfap2b, Sox10, and Msx1. I was wondering how the authors would interpret these results, given the data showing that these genes are thought to be necessary for neural crest migration, and differentiation. Would the transcript levels present in these cells be sufficient for these processes to occur?

[Editors’ note: what now follows is the decision letter after the authors submitted for further consideration.]

Thank you for submitting your article "Single-cell transcriptome analysis of neural crest migration reveals signatures of invasion and molecular transitions" for consideration by *eLife*. Your article has been reviewed by two peer reviewers, and the evaluation has been overseen by Marianne Bronner as the Senior and Reviewing Editor. The reviewers have opted to remain anonymous.

The reviewers have discussed the reviews with one another and the Reviewing Editor has drafted this decision to help you prepare a revised submission. Although the essential changes are summarized below, the full reviews are included for your reference.

Summary:

In the present manuscript, Morrison and colleagues conduct a single-cell transcriptome analysis of the R4 neural crest to investigate cell heterogeneity during collective cell migration. The study builds upon previous work from this lab that has shown that a subset of neural crest cells (trailblazers) guides migration and is molecularly distinct from the rest of the migratory stream. The authors (1) perform bulk RNA-seq experiments to identify transcripts enriched and depleted in the invasive migratory front; (2) survey the transcriptome of single neural crest cells to establish the molecular signatures of distinct subpopulations within the stream; (3) characterize a transcriptional signature of trailblazer cells; (4) conduct perturbation assays to investigate how trailblazer genes affect neural crest migration; (5) analyze cell heterogeneity within leading and following cells.

Essential revisions:

1) Comparison of the bulk seq results presented in Figure 1 with the other datasets.

2) More robust message from the seq analysis from which one could envision a "logic" for the synexpression of certain genes/pathways in trailblazers. Figure 6 does not show restricted expression in Trailblazers – are these the best examples?

3) Provide RNAscope quantification to validate their Trailblazer identity.

4) Perform more RNAscope analysis on their possible highest confidence targets.

5) Provide 3D reconstruction of streams from RNAscope?

6) The mixed expression in leading edge and in more caudal regions of the stream needs to be pointed in Results and explained in the Discussion.

7) Morpholino experiments need to include controls that show loss of the target protein.

8) To address the modest effects from the morpholinos, compare MO targeted vs. MO untargeted HNK1 positive cells in streams. Perhaps some untargeted cells can continue to perform as Trailblazers, and the MO-targeted cells then become followers… explaining the somewhat limited effects. This could be easily addressed by following the label on the MOs and showing the difference against HNK1. One would expect that MO. negative HNK1+ cells would be in front of MO+ cells. Also this might enable a much better quantification of the effects.

9) It is crucial to match RNAscope and MO-analysis. MO analysis on all their RNAscope samples and/or RNAscope on all their MO samples. If they are doing new RNAscope and or MOs, this "matching RNAscope and MO analysis" argument must be addressed.

10) Quantification of RNAscope data to confirm positive cells are indeed in the leading edge of the stream.

11) Please add a Discussion paragraph addressing differences between their previous qPCR study and this work.

*Reviewer #1:*

This study has technical advances of potential high relevance for the NC and developmental biology communities. The research presented focuses on single cell RNA sequencing to identify profiles of migrating NC cells. Specifically directed to pinpoint differences between single cells along the migratory streams that have been proposed to occupy significant positions as front runners or trailblazers, immediate followers and more distant trackers. This type of work could identify the specific contribution of certain genes to the proposed actions needed to perform as a Trailblazer, the cells that open the road for followers. Perhaps it could identify distinct required functions enabling cells to simply follow more advanced cells. While this could certainly be a significant message for developmental biologists at large, the work presented does not reach these targets.

Here the authors have done few changes and while more functional analysis seem essential, they instead provide minimal evidence of either restricted expression, or requirement for a specific function during migration.

In this new version, the authors added one set of image analysis for mRNA levels within a immunodetected HNK1 positive migratory stream. While this particular technique could be of great use, it really does not provide clear confirmation of the status assigned to the genes tested since their expression is actually apparent at multiple distinct positions in the stream and not restricted or graded according to the expected Trailblazer status designated through the mRNA seq analysis. The true value should have been the identification of key genes expressed in a pattern associated with specific position according to their migration "status", as Front, Lead and Trail, or Trailblazers or leading edge, vs. less advanced cells. This validation is missing. While the authors describe a quantification of mRNA spots on HN1 labeled r4-strem cells. The data provided is not that clear. In fact, the images provided suggest that the genes identified and selected through the single cell RNAseq do not provide a restricted expression to the expected positions.

As critical as the expression profile or perhaps even more, it would have been the identification of the functions provided from such genes. To this end the authors transferred some of the supplementary data from their original version, with identical images, to the main Results section of the revised manuscript. And tise set of KD results shows modest effects on the migration of neural crest cells at best. The evidence does not suggest a robust correlation towards a specific migratory status or position. While there are multiple genes in the table provided with apparent higher correlation to the Trailblazer, they only tried Wnt5B from those.

Similarly, while they identify two interesting groups or sets from the stage 11 embryos, which seem pretty interesting, again, no validation via expression is provided, nor a functional assessment.

Finally, while they have made an effort to compare their new data, to their 2015 data from QPCR, a clear message broadly assessing the differences, and commonalities, is still missing. Even more, no clear outcome of the analysis is evident. It is difficult to pinpoint an eloquent conclusion or message.

*Reviewer #2:*

As expected, the study is rich in transcriptomic data – the authors sequenced hundreds of cells of chick embryos from different stages. The results are properly organized and presented, and despite the abundance of heat maps and bar graphs the story flows well. The study is not without limitations – single cell transcriptomes are inherently noisy and fully distinguishing technical noise from true biological heterogeneity is challenging, if not impossible. But taken together, I believe the results represent a significant contribution. Indeed, this study perhaps makes the most compelling case for the existence/importance of Trailblazing cells in the migratory neural crest.

Of particular note is how the scRNA-seq results relate to the in vivo development of the neural crest. By comparing native cells to neural crest cultured in vitro, the authors show that significant transcriptional changes emerge with cell culture. Furthermore, the authors also attempt to bring some of the scRNA-seq findings back to the embryo, by conducting a number of in vivo experiments. They map the expression of some of the trailblazing genes by RNAscope/FRUIT, and subsequently they also show that a subset of those is required for proper neural crest migration. The first experiment produced specially compelling data – one is able to visualize individual Trailblazer cells within the migratory stream. The latter is potentially interesting but in my view needs additional controls.

[Editors' note: further revisions were requested prior to acceptance, as described below.]

Thank you for submitting your article "Single-cell transcriptome analysis of neural crest migration reveals signatures of invasion and molecular transitions" for consideration by *eLife*. Your article has been reviewed by two peer reviewers, and the evaluation has been overseen by Marianne Bronner as the Senior and Reviewing Editor. The reviewers have opted to remain anonymous. The consensus opinion is that your paper could be acceptable as a Tools and Resources (TR) paper with some additional validation as discussed below. Our sense is that in order to qualify as a Research Article (RA), you would need to perform significant additional functional tests that may take quite a bit of time to complete.

Summary:

The consensus opinion is that the manuscript has been improved by the additional of functional data but it remains critical to show that a "Trailblazer" signature is actually unique to the leading cells in order to support the authors claims.

Essential revisions:

The full reviews of the reviewers are included below for the authors' reference. Both reviewers feel that further validation of the Trailblazer signature is necessary. Thus, it is critical that you do one of the following, with the first suggestion best in line for a TR paper:

1) Identify 5-10 genes that are either restricted at the front, or have graded expression with highest intensity towards the leading edge; and/or

2) Perform additional functional experiments to see if any of the identified genes is functionally important with a major phenotype.

Please also note that you should include better controls for the morpholino experiments.

I hope you find the reviewers comments helpful in revising the paper.

*Reviewer #1:*

This revised version has added changes in a short time, and the authors have put effort in amending the previous version. However, the main objection to this study has been the biological evidence for the significance of the transcriptomic signatures suggested for the trailblazers, the foremost advanced migratory crest cells leading the way for follower neural crest cells. Unfortunately, the evidence presented seems insufficient.

Key to the significance is Figure 6, looking at expression and function. Expression profiles were monitored through quantifiable in situ hybridization (RNAscope) and co-immunostaining with HNK1 to reveal the full migratory stream. The results are intriguing, because while there are clearly some cells near the front with high expression, there is no apparent restriction to the leading cells. Instead each gene presented shows a) expression in some (not all, not even the majority) cells near the edge being very strong, and b) multiple patterns, with some displaying strong expression at the opposite end of the stream (back delayed followers), or leading edge or just more anterior cells within the stream with intermediate levels of expression. These results offer a complex pattern not easily ascribed to a restricted subset of cells in a particular level. While the quantification provided may suggest that there is a pattern, their analysis is limited to a restricted number of cells, nearby their ideal high-expressing target. We previously asked, are these the best candidates? This being the third version of the manuscript, may suggest so.

Then they assess the function of some target genes in the migration. They do not address exactly the same monitored for expression (Figure 6) which would have been a nice demonstration had they observed a clear phenotype even if the expression was not optimal. They suggest to have used the same criteria carefully laid out for the expression analysis, but do not restrict themselves to the top Trailblazer signature and instead they added targets based on migration-likelihood as criteria. If these genes are expressed in a pattern to support the Trailblazer claim, that would be perfect, but no additional expression evidence is presented. But the most critical aspect is simply that the Morpholino treatment of any of these genes, do not provide a strong phenotype. The average migration effect seen by HNK1 in treated streams only displays a reduction of 10%! Control Mo displays a range around 96% migration, their best candidate Wnt5b, has an average of 91% and the next two best candidates GATA5 and LUM show about 94% migration. This does not seem to be a robust phenotype. They suggest that this is due to migration from wt, cells that did not received the morpholinos, but no evidence supports this claim. The results presented in the main Figure 6 do not provide a robust effect. Few of their electroporations leads to a reduced expression compared to the control range, their top candidate Wnt5b being perhaps 15% reduced over the control. It would have been good to show the images of that specific KD, so even if it is limited at least people could see it. The images for panel G provide a confusing picture. IN one hand, EDN1 and PKP2 MO treated cells appear near the edge of the stream…

*Reviewer #2:*

In the revised version of manuscript, Morrison and colleagues conduct an extensive single-cell transcriptome analysis to investigate cell heterogeneity and Trailblazer cell identity in neural crest cells from the R4 migratory stream. The results reveal a transcriptional signature that remains essentially unchanged in cells isolated from embryos in distinct stages of development (HH11, HH13 and HH15) (Figure 4). The authors postulate that this signature would define the identity of the Trailblazer cells, which are located at the migratory front and have a crucial role in the establishment of the migratory stream. The unbiased identification of a subpopulation of neural crest cells through scRNA-seq is an important result; however, as highlighted in previews comments, it is crucial that such findings be validated in vivo.

This reviewer finds that the new version of the manuscript does not unequivocally show that Trailblazer cells express the genes that are part of their 'transcriptional signature'. In Figure 6, the authors employ RNAscope to visualize and quantify the expression of the genes mostly enriched in the Trailblazer signature. The results are difficult to interpret, as the cells that have the highest expression of the genes are in many cases far away from the leading edge (Desmin). In another example, the target gene seems to be downregulated in the migratory front (Tescalcin). I found the figure confusing, as the authors opt to quantify transcript levels in the cells that have the highest expression of the transcript, rather than focusing on the cells at the leading edge (the request from the reviewers was RNAscope quantification in Trailblazer cells). Thus, while I agree that this data shows neural crest heterogeneity, I don't see how it supports the hypothesis of a Trailblazer transcriptional signature.

I acknowledge the authors' comment on how the scRNA-seq delineates a complex trailblazing signature, composed of hundreds of genes, but I don't see how this this should hinder in vivo validation of the transcriptome data. Figure 4 clearly shows a group of genes that are enriched in the trailblazing cells and depleted in other populations. If the analysis and the hypothesis are correct, one would expect that expression mapping would show higher levels and co-expression of these genes in Trailblazers. It is this reviewer's opinion that in vivo validation of RNA-seq – especially in the case of noisy single cell data – should be emphasized. Additionally, the lack of consistency in the genes that are highlighted in the bioinformatic analysis, the genes that are analyzed with RNAscope and the genes that are chosen for loss of function analysis raises questions about the dataset. I have no problem with the authors focusing in a particular subset of genes for the in vivo experiments, but a certain level of consistency would make the manuscript easier to follow.

As previously stated, the scRNA-seq analysis performed by the authors represents an important resource for community, yielding an interesting hypothesis. However, the fact that the link between the scRNA-seq results and the in vivo analysis is tenuous creates a disconnect between the two parts of the manuscript. This should be resolved for improvement on the consistency and reliability of the study.

[Editors' note: further revisions were requested prior to acceptance, as described below.]

Thank you for submitting your article "Single-cell transcriptome analysis of neural crest migration reveals signatures of invasion and molecular transitions" for consideration by *eLife*. Your article has been reviewed by two peer reviewers, and the evaluation has been overseen by a Marianne Bronner as the Senior and Reviewing Editor. The reviewers have opted to remain anonymous.

The reviewers have discussed the reviews with one another and the Reviewing Editor has drafted this decision to help you prepare a revised submission. I have attached the full reviews of both reviewers which are highly concordant. While they feel that the data are valuable as a Tools and Resource paper, both reviewers felt that the way the manuscript was presented was confusing and that it would be important to rewrite the manuscript in a manner that doesn't over interpret the results. I think that you would be able to do this with textual revisions alone but would urge you to spend particular care on addressing the points raised in the full reviews, as both reviewers commented that they felt you had not addressed their earlier concerns and were trying too hard to put your conclusions in a positive light rather than addressing the caveats of the single cell RNA-seq analysis and the difficulties of linking this to in vivo expression analysis and function.

Given the history of this manuscript, this must be your last opportunity to produce an acceptable version.

Essential revisions:

1) The authors should aim to provide a clear, and accurate description of their findings. These in vivo findings need to be spelled clearly: These genes were identified by scRNAseq and bioinformatics (through a stringent protocol with harsh significance in mind), as specific to the expected Trailblazer at the narrow front edge of the migratory front, however, they do not seem to appear restricted in that fashion in vivo, and insteada) None of the 13 genes tested so far (which include some of the top candidates) are expressed in all of, or the majority of the cells at the front edge.b) some may appear in a few or some but not in all of the cells (~30% or 1/3?) of the (narrow) front migratory edge.c) some of the genes may display salt and pepper patterns which may include strong expression in a few cells in the narrow front edge and similarly strong signal in some cells in the beginning or middle portions of the stream (not at the front edge of migration).d) Some of the genes where not found at the front edge at all, and instead appear to be expressed more strongly at various positions within the migratory stream.

2) The authors should provide a revised version of the so called "complex profile/signature of expression for a Trailblazer" If the genes are not specifically expressed in all of them, or the majority, and if these genes may be expressed in the stream, then what is the so called complex signature of the trail blazer? (Not to mention its relevance…) The authors could offer some useful insights regarding the limitations or caveats of the scRNAseq at the current time/or in these complex settings. Perhaps cumulative RNA vs. freshly made RNA.

3) The novel RNAscope data shows heterogeneous expression of genes that are part of the "Trailblazer signature". While this reviewer finds that a thorough transcriptomic characterization of single RNA cells may be a useful resource for the community, the results of the in vivo analysis require further clarification.

4) Upon presenting the expanded RNAscope analysis, the authors conclude that "the combination of fluorescence in situ hybridization with IHC and tissue clearing allowed the confirmation of a small subset of the Trailblazer transcriptional signature". This statement seems to imply that heterogeneous expression is sufficient to validate transcripts as being part of the Trailblazer signature. According the Introduction of the manuscript such genes should be expressed by cells "narrowly confined by the migratory front". This discrepancy needs to be addressed or the reader will not be able to link the transcriptomic data with the in vivo expression analysis.

*Reviewer #1:*

The Authors previously used cell sorting and q-RT-PCR to identify a unique molecular signature of 16 out of 96 genes that was stable and consistent in cranial neural crest cells narrowly confined to the most invasive front of the migratory stream that we termed 'Trailblazers' (McLennan et al., 2015). In this manuscript, they have further scrutinized the expression profile of individual cells to address molecular changes and singularities that could help us better understand the behavior of migrating cells, specifically those at the fore front of the migration. The appealing idea being that these front runners might be endowed with unique capabilities to determine direction, remove ECM, perhaps even push other cells to the side, open a path, and lay the road and or instructions for follower cells.

There is a lot of value in this hard and fine work. They had accomplished a technically challenging task addressing the expression profiles of migrating neural crest cells. They provide significant information regarding the differences between in vitro cultured and in vivo migrating neural crest cells. They confirm some of the genes they previously identified as enriched in the "Trailblazer" cells using a single cell RNA seq approach. Yet the outcome of this huge endeavor in its present format, is misleading. As presented by the authors, one would assume that they have identified a unique "expression signature" of Trailblazer cells, which migrate at the front edge of the migratory stream in chick embryos, which they define as "narrowly confined by the migratory front". The implication being that a few genes would be expressed in a specific pattern associated with the migratory front edge. Does this mean a strong expression of a few genes consistent at the front edge? This argument is made throughout the paper in multiple places. But they do not provide the evidence necessary to grant such claim, and should be corrected throughout.

Key to these arguments would be a clear, measurable, objective definition of the front edge. In terms of its distance from the dorsal neural tube or departure point, truly the edge, near the edge, 2, 3 5 cells behind the apparent edge? And then the width or range of the front edge, all the cells visible at the edge? Some angle or curvature to restrict it? And what is the volume of the front edge? The use of ill-defined, subjective terms provides ample room for misinterpretation and confusion.

If we were to think of our arm and hand as a migratory stream departing from our trunk, what could be considered as the Trailblazers, the nails? The tip of the thumb? Or the index fingers only, the distal, middle or proximal portions of some or all of the phalanges? Would it be enough to look at the back of the hand/dorsal portion of the phalanges, or the palm portion/ ventral, or the middle? etc…

While performing the transcriptomic portion of the work, a lot of effort was placed to define and collect cells from specific regions of the stream to be analyzed. Yet upon validation of expression in embryos, a much laxer approach is taken and confusion ensues.

The heading of one of the Results sections reads: "Trailblazer cells are identified by an ensemble of genes", and after presenting the results in cryptic and limited format, concludes: "Thus, the combination of fluorescence in situ hybridization with IHC and tissue clearing allowed the confirmation of a small subset of the Trailblazer transcriptional signature." Yet the expression of the genes here presented is not restricted or biased to the front edge.

The authors perform single cell transcriptomics and come up with a selected list of genes thought to decipher the molecular signature of the "Trailblazer" cell. They try to corroborate their thorough and delicate transcriptomics and bioinformatics work via two assessments. The first one consisting of expression analysis, and the second on Morpholino mediated knockdown. The knockdown provided minimal effects at best and thus could not confirm the tenants of a "relevant" trailblazer expression. Perhaps combinatorial approaches could reveal function, but at this point no more evidence is provided.

To demonstrate expression profile associated with Trailblazer character, the authors performed a nice combination of immunofluorescence to identify all the migratory NC via HNK1, combined with quantifiable in situ hybridization analysis, "RNAscope". The criteria provided to select genes to prove and validate the trailblazer character states:

1) genes with enriched expression in Trailblazer cells;

2) high percentage of single Trailblazer cells in which the gene was detected and;

3) high level at which the gene was expressed by the Trailblazer cells.

(The genes tested are PKP2, DESMIN, KAZALD1, TESCALCIN, TROPONIN, ANEXIN, BAMBI, GLYPICAN3, DESMOPLAKIN, BDNF, TFPI2, NEXILIN, CDH5).

In no case, a unique pattern is presented that strictly labels with the highest signal the expected trailblazer cells at the front edge of the migratory stream defined as "narrowly confined by the migratory front". Instead, in the best cases for their argument (PKP2, and KAZALD1), expression seem to appear strong or strongest in a few of the visible trailblazer cells. These genes do not appear consistently in their strongest expression profile at the front edge, in all the trailblazers. Some Trailblazer cells clearly do not express these genes and or express them weaker than follower cells. And in multiple cases stream or follower cells display a wide range of expression levels varying from the strongest to no expression. And one has to wonder what is the depth of the image presented? Does this represent the whole edge, or would one imagine 3 similar projections to cover the volume of the front edge? If so, what is the number of cells at the front edge that strongly express the interrogated marker? Further, when they analyze the expression relative to neighbors, the effort is limited to very few cells, when the same arguments made for distance to the edge, width and depth should apply.

This is even more clear when analyzing the other 11 genes they present. Desmin, and Troponin1, display 1 and 2 cells respectively, with the strongest expression (red) far behind the expected Trailblazers (are not at the very front) and instead are behind multiple cells with 2/3 or 1/2 as much expression. And unfortunately, the evidence provided for their analysis of the additional other 8 genes, does not support their case either, and actually are not a better example of what one would have expected for a trailblazer profile of expression.

Importantly, for some undisclosed reason, they do not include the same targets on expression and function (only 3 of the 9 targets assessed via Morpholinos are also tested amongst the 13 targets used for expression analysis with RNSAscope).

Given the lack of evidence of relevant role for the genes tested, and that no signature is confirmed by expression in the embryo, the current version would be a misleading tool or resource.

The authors should aim to provide a clear, and accurate description of their findings. These in vivo findings need to be spelled clearly:

These genes were identified by scRNAseq and bioinformatics (through a stringent protocol with harsh significance in mind), as specific to the expected Trailblazer at the narrow front edge of the migratory front, however, they do not seem to appear restricted in that fashion in vivo, and insteada) None of the 13 genes tested so far (which include some of the top candidates) are expressed in all of, or the majority of the cells at the front edge.b) some may appear in a few or some but not in all of the cells (~30% or 1/3?) of the (narrow) front migratory edge.c) some of the genes may display salt and pepper patterns which may include strong expression in a few cells in the narrow front edge and similarly strong signal in some cells in the beginning or middle portions of the stream (not at the front edge of migration).d) some of the genes where not found at the front edge at all, and instead appear to be expressed more strongly at various positions within the migratory stream.

The authors should provide a revised version of the so called "complex profile/signature of expression for a Trailblazer" If the genes are not specifically expressed in all of them, or the majority, and if these genes may be expressed in the stream, then what is the so called complex signature of the trail blazer? (Not to mention its relevance…) The authors could offer some useful insights regarding the limitations or caveats of the scRNAseq at the current time/or in these complex settings. Perhaps cumulative RNA vs freshly made RNA…

*Reviewer #2:*

This revised version of the manuscript by Morrisson and colleagues on scRNA-seq analysis of neural crest cells includes an expanded RNAscope analysis of putative Trailblazer genes. Reviews of the previous versions of the manuscript emphasize the need for further in vivo validation of the transcriptomic datasets. The novel RNAscope data shows heterogeneous expression of genes that are part of the "Trailbrazer signature". While this reviewer finds that a thorough transcriptomic characterization of single RNA cells may be a useful resource for the community, the results of the in vivo analysis require further clarification.

Upon presenting the expanded RNAscope analysis, the authors conclude that "the combination of fluorescence in situ hybridization with IHC and tissue clearing allowed the confirmation of a small subset of the Trailblazer transcriptional signature". This statement seems to imply that heterogeneous expression is sufficient to validate transcripts as being part of the Trailblazer signature. According to the Introduction of the manuscript such genes should be expressed by cells "narrowly confined by the migratory front". This discrepancy needs to be addressed or the reader will not be able to link the transcriptomic data with the in vivo expression analysis.

---

## [Author Response]

[Editors’ note: the author responses to the first round of peer review follow.]

Reviewer #1:[…] Results section 1• Why were these two stages being chosen? Stage 13 appears to be very close to stage 15 in terms of gene expression. Bioinformatics analyses also did not appear to justify there was a stage specific difference.

We agree and have clarified this in the text. Stage 11, 13 and 15 are separated developmentally by approximately 8 hours. Stage 11 coincides with the initiation of migration from the neural tube, Stage 13 is active migration and Stage 15 is colonization of the target tissue. Only after doing these experiments did we discover the similarity between Stage 13 and 15.

• At what stage did the authors label the premigratory neural crest cells?

We have clarified this in the text.

• There was no description whether the 16 genes identified in McLennan et al., 2015 were found in the bulk RNA seq and what were the new players. For example the analyses should have been performed on genes such as, Hand2/GPC3 as invasive front signature, PDGFRL/CXCR1 as stream signature, and EPHA4 and FOXD3 as trailing signature identified in McLennan et al. 2015. These need to be first established in the bulk RNA seq analysis.

We have clarified this in the text and in a figure. The complete differential expression analysis of the bulk subpopulations (both validating our previous findings and establishing new signatures) may be found in Supplementary file 1.

Results section 2• "we isolated and profiled individual cells from different stream positions at three developmental stages". “We subdivided neural crest cell migratory streams at HHSt13 and HHSt15 into three spatially distinct, non-overlapping subregions by careful manual dissection and single cells were isolated by FACS". The authors need to justify the rationales behind the selection of stages (two versus three) and dissection methods (front/trail vs. front/lead/trail) for the bulk and single cell RNA seq. Were there other previous experiments and studies that the authors or others had performed that would help justify the differences?

We have expanded our rationale in the text.

• "Hierarchical clustering of single neural crest cell averages….." The initial analysis should have been performed without averaging the single cell transcriptome values according to their migration positions. The current experimental design, by clustering single cell samples, could have unavoidably biased the results. The conclusion reached also did not add any new insight into the study. Figure 2 could be removed. The aim of the experiment was to identify new genes/signatures that can define the different subpopulations and the authors could have used the previous 16 genes (McLennan et al., 2015) or other known targets from the bulk RNA seq in Figure 1 or other studies as a guide to distinguish the front cells.

We have moved Figure 2 (bulk RNA-seq averages) to the supplementary material. Although either the RT-qPCR or bulk RNA-seq signatures could have been used to identify new genes/signatures to define cell subpopulations, these data were constrained by the manual cuts to subdivide the stream. Instead, we chose an unbiased approach to identify novel cell subpopulations independent of the position of the manual dissections to subdivide the stream and have clarified this in the text.

Results section 3• "in vitro transcriptional signatures are distinct from any in vivo profile". This section appears not relevant to the rest of the study. This section does not give any new insight into the cellular heterogeneity of migrating neural crest cells.

We appreciate the reviewer’s suggestion. Since discrete neural crest cell migratory streams are not re-capitulated in vitro, but in vitro assays are readily used for cell migration studies, we wished to confirm our hypothesis and previous RT-qPCR data (McLennan et al., 2015) that in vitro transcriptional signatures are very distinct from our in vivo results and is useful information. Thus, further supporting the model that in vivo neural crest microenvironmental signals play an important role in establishing in vivo transcriptional signatures. We have moved the in vitro data to later in the manuscript and clarified this in the text.

Rest of the Results sections• “Our previous RT-qPCR (McLennan et al., 2015) and bulk RNA-seq presented above (Figure 1) show that cranial neural crest cells at the invasive front of a stream have aspects of their transcriptome that are conserved during migration.” This rationale does not fit into the experimental design for Figure 5. If the invasive front were to have a conserved transcriptome during migration, then all cells should have been clustered for analysis. The experiment presented in Figure 5 instead appears to try to identify heterogeneity of the front cells dissected from the embryos.

We have clarified this in the text.

• Figure 5: Where did the front cells for stage 11 come from? It was stated in the first paragraph of the subsection “Single-cell RNA-seq identifies gene expression variances based upon spatial position within the neural crest cell stream and temporal progression along the migratory pathway” that there was no subdivision of stage 11 cells.

We refer to HHSt11 cells as FRONT, since the small number of migrating neural crest cells adjacent to r4 does not allow for subdivision of cells into subgroups. We have clarified this in the text.

• Use different colors for Trail and Trailblazers in Figure 6.

Done.

• Subsection “The scRNA-seq Trailblazer signature is uniquely defined and associated with neural crest cells confined to the invasive front of the stream”, last paragraph: It is interesting that the authors were able to identify previously known pathways, such as the VEGF signaling, that were associated with Trailblazer identity. However, as the purpose was to discover new pathways, an unbiased search should be performed instead of a focus on developmental relevant signaling pathways. As noted, a significantly higher number of pathways were found to be upregulated and many of the p-values were relatively high (Figure 6) suggesting that there might exist other "developmentally irrelevant" pathways that were associated with trailblazer signatures.

This is an excellent suggestion and we have added lists of all statistically significant pathways for all single cell RNA-seq subpopulations discussed in Supplementary file 6.

• Please highlight the p-value in Figure 6 that represent significance difference.

Done.

• Figure 6 can be better presented with the p-values align according to their values from top to bottom.

We appreciate the reviewer’s suggestion but wished to highlight the links to the VEGF and axonal guidance signaling pathways and thus plotted by alphabetical order. P values can still be easily seen in the graph.

• What were the criteria to select the gene list for presentation in Figure 6? Were these genes most highly enriched or depleted?

We selected genes that we considered of general interest to the neural crest and embryonic cell migration fields. The most enriched and depleted genes may be found in the differential gene lists contained in Supplementary file 4.

• Highlight in the text and figure legend for Figure 6 that those genes highlighted in red were previously found in McLennan et al., 2015.

Done.

• The conclusion "These results highlight the upregulation of many genes within the trailblazer subpopulation, supporting the hypothesis that trailblazers are a dramatically active subpopulation of cells at the most invasive front during neural crest migration" needs to be substantiated (and if possible reworded to omit superlatives.

We agree and have reworded the text.

• " We also found that CDH7, CXCR4 and EPHB1 were upregulated in the HHSt11 single cells (Figure 6)." What were these single cells? Were there any bias in stage 11 clusters 1 and 2 identified earlier?

We have included text within the manuscript to describe these results more accurately and have added the differential expression analysis between HHSt11 Cluster 1 and 2 to Supplementary file 4.

• Subsection “The scRNA-seq analysis validates some of our original 16 gene RT-qPCR Trailblazer signature”: the authors said there was "no significant enrichment" for a number of genes found in previous studies meaning the expression of these genes were upregulated but not statistically significant or were downregulated instead? Following up on this, if these genes were not upregulated in the trailblazer population, were they found to be upregulated in any of the subpopulations of the migrating neural crest cells?

We agree and have clarified this in the text.

• Subsection “Identification of two distinct transcriptional signatures associated with neural crest cells isolated shortly after dorsal neural tube exit”, second paragraph: How was the GO search performed for clusters 1 and 2? Why were there no depleted and enriched terms for clusters 1 and 2 respectively?

The Gene Ontology (GO) analysis was completed using the R package ClusterProfiler (3.2.6). GO terms with a p value less than 0.05 and a q value below 0.1 were considered significant. Significant GO terms were condensed using the simplify function with an adjusted p value cutoff of 0.05. While the results are surprising, there simply were no depleted or enriched terms because no pathways contained enough genes from our differential expression analysis to be statistically significant. We have clarified this in the text.

• Please correct typos. “To identify signaling pathways enriched within HHSt11 clusters, we performed pathway enrichment analyses "for" the lists of genes differentially expressed by HHSt11 Cluster1 and Cluster2 versus all other in vivo NC cells to[…]”.

Done.

• "There may be even more dramatic differences between the pathways associated with each cluster in the form of significance." It is not clear what was the message the authors were trying to convey in this sentence.

We have clarified this in the text.

• Subsection “Identification of two distinct transcriptional signatures associated with neural crest cells isolated shortly after dorsal neural tube exit”, last paragraph: It is intriguing to find that WNT/PCP signaling were respectively up and downregulated in clusters 1 and 2. It might shed important functional roles of these distinct subpopulations of stage 11 neural crest cells by altering WNT at this stage and to see if and how skewed production of these subpopulations lead to phenotypes in neural crest cell migration and formation of the Trailblazer population at later stages.

We agree that the role of WNT/PCP signaling deserves future investigation. We have added a list of genes differentially expressed between HHSt11 Cluster 1 and 2 to Supplementary file 4 and highlighted genes associated with WNT/Β-Catenin and PCP signaling pathways.

• "This conclusion is most likely the result of increased heterogeneity of these subpopulations due to the increased size when compared to the other subpopulations examined." The opposite appears to be true to common sense: reduced sample size will likely lead to increased heterogeneity. The findings presented here suggest that trail and lead subpopulations were highly heterogeneous and no defined signatures can be identified. Increasing the sample size (number of cells) in future experiments on the other hand might help to identify small subpopulations.

We have clarified this in the text. The measurement of a population’s heterogeneity is based upon both the sample size (as the reviewer notes) as well as the actual heterogeneity of the population. While similar numbers of cells were analyzed from each subpopulation (50-60 cells/each; Figure 2), the total numbers of Leader and Trailer cells within the stream is much larger than the number of Front cells.Additionally, McLennan 2015 showed regional diversity throughout the stream, thus we would expect a larger subpopulation of the stream (Trailers, 70% of the stream) would have more heterogeneity than a smaller subpopulation (Invasive Front, 5% of the stream). Less statistically significant differential expression is likely to occur when larger subpopulations with more heterogeneity are averaged.

• The result description for Figure 6—figure supplement 1 should be elaborated and should be placed within the Results section instead of in the Discussion section.

Done.

Reviewer #2: […] I found the manuscript to be well structured, given the challenge of condensing and organizing transcriptomic data from hundreds of cells. This is partly due to the decision (a good one, in my view) of focusing specifically on the differences between leading vs. trailing cells. Indeed, the manuscript represents a substantial addition to the work the group has done on this topic – while also resulting in a useful resource for future studies. Nevertheless, I believe it would greatly benefit from additional in vivo analysis. While reading the paper, I found myself trying to visualize how the distinct populations characterized through transcriptomics were distributed within the R4 stream. Where would the Trailblazer cells be and how many of them would be present? Mapping the expression of the novel trailblazer genes back into the embryo would be an important validation of the analysis, and yield in spatial information that could clarify the molecular heterogeneity observed.1) How do the authors reconcile the results from the bulk to the single cell experiments? I would expect to find the genes that are part of the Trailblazer molecular signature to be enriched in the invasive front, but that is only the case for a handful of genes. Is this a matter of sensitivity / sequencing depth?

We agree and appreciate the suggestion to clarify this in the text. Briefly, single cell RNA-seq identified Trailblazers as a subset of the Front populations (8, 50 and 32% at HHSt11, 13 and 15, respectively). Therefore, when entire Front populations are averaged, as in the bulk analyses, some of the Trailblazer genes may or may not be statistically significant.

2) The use of single-cell transcriptomics allowed the authors to refine their previous findings on the molecular signature of the trailblazer cells. Yet I found the discussion on the significance of this signature to be somewhat vague. Are Trailblazer cells characterized by the co-expression of these genes? Are there any Trailblazer-specific individual factors? How heterogeneous is the Trailblazer population? In short – if one needs to identify Trailblazer cells, what are the factors they should be focusing on?

We have clarified this in the text.Briefly, our scRNAseq analysis did not reveal any individual gene that is able to distinguish the Trailblazers from the remaining migratory stream. Rather, we assert that Trailblazers are characterized by a large number of differentially expressed genes (added in Supplementary file 4).

3) While I appreciate the authors' use of microdissection to incorporate a degree of spatial resolution in the analysis, the work would greatly benefit from additional in vivo experiments. The authors should confirm co-localization of at least some of the transcripts that are part of the trailblazer molecular signature in embryonic tissue. Is it possible to map individual trailblazer cells utilizing these markers? How large is the population of trailblazers in the stages analyzed, and is this consistent with the results of the single cell transcriptome data?

We agree and have selected a subset of genes to confirm expression based on the criteria: 1) genes with enriched expression in Trailblazers, 2) the percentage of single Trailblazer cells in which the gene is detected, and 3) the level at which the gene is expressed by the Trailblazers.This is presented in a revised Figure 7 and expanded in the Materials and methods section.

4) The morpholino experiments on Figure 6 come across as an afterthought. Have these morpholinos been published (I do not see the morpholino sequences in the method sections). How are these genes expressed? Proper controls for efficiency and specificity should also be included.

We agree and have included the additional data. We have expanded our investigation of Trailblazer function via morpholinos and have included the data in Figure 7 (Morpholino sequences are included in Figure 6—source data 1).

5) Surprisingly, Trailblazing cells have weaker expression of genes that are thought to be central to the neural crest regulatory program such as Tfap2b, Sox10, and Msx1. I was wondering how the authors would interpret these results, given the data showing that these genes are thought to be necessary for neural crest migration, and differentiation. Would the transcript levels present in these cells be sufficient for these processes to occur?

Sox10 for example, is highest in the most recently delaminated neural crest cells and its expression decreases over developmental time. That is, by HH St15, the front of the stream is no longer Sox10 positive. We have seen this in qPCR, RNAseq and in situ approaches over the years. We see a similar expression profile for FoxD3. We have not done in depth expression analysis of other known neural crest genes such as TFAP2B or MSX1, but they are mainly known to be involved in the specification of neural crest, not necessarily the active invasion of unexplored microenvironments. In this study, both TFAP2B and MSX1 are expressed higher at Stage 11 that at the later stages, consistent with their roles in the neural crest cell gene regulatory network (Supplementary file 2). Furthermore, this study did not compare neural crest cells to other cell types and therefore genes that are reasonably homogeneously expressed by neural crest cells will not be highlighted.

[Editors' note: the author responses to the re-review follow.]

Essential revisions:1) Comparison of the bulk seq results presented in Figure 1 with the other datasets.

We have included the requested comparisons. Supplementary file 1 compares differentially expressed genes between the Front and Stream by bulk RNAseq and highlights the presence of genes from the 16 gene scRT-qPCR Trailblazer signature. Supplementary file 4 highlights the presence of the enriched and reduced genes from Figure 1 in the lists of genes differentially expressed by Trailblazers from scRNA-seq. We have included text in the Results section to describe the comparisons.

2) More robust message from the seq analysis from which one could envision a "logic" for the synexpression of certain genes/pathways in trailblazers. Figure 6 does not show restricted expression in Trailblazers – are these the best examples?

Done.

3) Provide RNAscope quantification to validate their Trailblazer identity.

Done (revised Figure 6).

4) Perform more RNAscope analysis on their possible highest confidence targets.

We agree with the reviewer that the validation of our Trailblazer transcriptional signature would enhance the value of our data by identifying genes and gene families for further functional analysis. To begin to address this, we confirmed that the Trailblazer transcriptional signature validated many of the genes we identified by RT-qPCR analysis (Supplementary file 1 and 4) to be expressed high and consistent over time (McLennan et al., 2015). Second, the Trailblazer transcriptional signature is defined as an *ensemble* of differentially expressed genes (900 enhanced, 400 reduced) from our PCA analysis and highlights the heterogeneity of expression within cells at the invasive front. To address the heterogeneity of expression, we confirmed that neighboring cells within the invasive front and stream may have dramatic differences in mRNA expression of an individual gene by using multiplexed fluorescence in-situ hybridization (RNAscope) and quantitative single cell analysis (revised Figure 6).

We completely understand the reviewer’s request to visualize the expression of an individual gene associated with the Trailblazer transcriptional signature, but suggest this may or may not uniquely identify a Trailblazer cell. For example, if Gene A is enhanced within the Trailblazer transcriptional signature *ensemble*, its expression may also be high within follower cells, but since the follower cell lacks the full complement of differentially expressed genes as part of the *ensemble*, it would not uniquely identify a Trailblazer cell. To clarify this, we have: (i) added quantitation and a 3D movie of our RNAscope data to strengthen the message of the heterogeneity of individual gene expression of Trailblazer transcriptional signature (revised Figure 6; Video 1); (ii) added the analysis of 3 other genes of the Trailblazer transcriptional signature and morpholino knockdown experiments (revised Figure 6 and Figure 6—figure supplement 2) and; (iii) edited the text to elucidate the ensemble aspect of the Trailblazer transcriptional signature.

5) Provide 3D reconstruction of streams from RNAscope?

Done.

6) The mixed expression in leading edge and in more caudal regions of the stream needs to be pointed in Results and explained in the Discussion.

Done.

7) Morpholino experiments need to include controls that show loss of the target protein.

We have included supplementary material that shows alterations in target mRNA in chick cells subjected to splice blocking morpholinos and added text to the Results and Methods sections (revised Figure 6—figure supplement 2), as we do not have antibodies to these genes to visualize protein levels.

8) To address the modest effects from the morpholinos, compare MO targeted vs. MO untargeted HNK1 positive cells in streams. Perhaps some untargeted cells can continue to perform as Trailblazers, and the MO-targeted cells then become followers… explaining the somewhat limited effects. This could be easily addressed by following the label on the MOs and showing the difference against HNK1. One would expect that MO. negative HNK1+ cells would be in front of MO+ cells. Also this might enable a much better quantification of the effects.

Done (revised Figure 6—figure supplement 3).

9) It is crucial to match RNAscope and MO-analysis. MO analysis on all their RNAscope samples and/or RNAscope on all their MO samples. If they are doing new RNAscope and or MOs, this "matching RNAscope and MO analysis" argument must be addressed.

For expression analysis by RNAscope, we selected genes for expression confirmation based on the criteria: (1) genes with enriched expression in Trailblazer cells; (2) high percentage of single Trailblazer cells in which the gene was detected and; (3) high level at which the gene was expressed by the Trailblazer cells. For knock down experiments, we selected 10 genes from distinct molecular families to test, based on the same criteria used for selecting genes for expression analysis, but also for predicted roles in neural crest migration. Thus, different rationales produced to divergent lists of genes. We have now added the expression analysis of 3 genes from the knock down list (revised Figure 6).

10) Quantification of RNAscope data to confirm positive cells are indeed in the leading edge of the stream.

See #3 and #4.

11) Please add a Discussion paragraph addressing differences between their previous qPCR study and this work.

Done.

[Editors' note: further revisions were requested prior to acceptance, as described below.]

Essential revisions:The full reviews of the reviewers are included below for the authors' reference. Both reviewers feel that further validation of the Trailblazer signature is necessary. Thus, it is critical that you do one of the following, with the first suggestion best in line for a TR paper:1) Identify 5-10 genes that are either restricted at the front, or have graded expression with highest intensity towards the leading edge; and/or2) Perform additional functional experiments to see if any of the identified genes is functionally important with a major phenotype.Please also note that you should include better controls for the morpholino experiments.

We have addressed the final Essential revision based on the reviewers’ request to add in situ data for several Trailblazer genes. We now include expression analysis of 6 more Trailblazer genes (for a total of 13) expressed in but not exclusive in these cells. We are very grateful for the reviewer’s helpful suggestions that have strengthened the manuscript.

[Editors' note: further revisions were requested prior to acceptance, as described below.]

Essential revisions:1) The authors should aim to provide a clear, and accurate description of their findings. These in vivo findings need to be spelled clearly: These genes were identified by scRNAseq and bioinformatics (through a stringent protocol with harsh significance in mind), as specific to the expected Trailblazer at the narrow front edge of the migratory front, however, they do not seem to appear restricted in that fashion in vivo, and insteada) None of the 13 genes tested so far (which include some of the top candidates) are expressed in all of, or the majority of the cells at the front edge.b) some may appear in a few or some but not in all of the cells (~30% or 1/3?) of the (narrow) front migratory edge.c) some of the genes may display salt and pepper patterns which may include strong expression in a few cells in the narrow front edge and similarly strong signal in some cells in the beginning or middle portions of the stream (not at the front edge of migration).d) Some of the genes where not found at the front edge at all, and instead appear to be expressed more strongly at various positions within the migratory stream.

We have revised the Results and Discussion sections to provide a clear and accurate description of our findings, including a more accurate description of the gene expression analysis and inconsistencies between the RNAseq and in vivo expression analysis data exactly as the reviewer’s describe above.

2) The authors should provide a revised version of the so called "complex profile/signature of expression for a Trailblazer" If the genes are not specifically expressed in all of them, or the majority, and if these genes may be expressed in the stream, then what is the so called complex signature of the trail blazer? (Not to mention its relevance…) The authors could offer some useful insights regarding the limitations or caveats of the scRNAseq at the current time/or in these complex settings. Perhaps cumulative RNA vs. freshly made RNA.

We have clarified this in the text and also provide a list of genes expressed in greater than 95% of trailblazer neural crest cells (Figure 5) and have clarified the modest changes in neural crest cell distance migrated after knockdown of several trailblazer genes, including the addition of Wnt5B MO and p-values for each of the morpholino experiments (Results section, Figure 6, and Figure 6 legend).

3) The novel RNAscope data shows heterogeneous expression of genes that are part of the "Trailbrazer signature". While this reviewer finds that a thorough transcriptomic characterization of single RNA cells may be a useful resource for the community, the results of the in vivo analysis require further clarification.

We appreciate the reviewer’s helpful suggestions and have revised the text to be clear.

4) Upon presenting the expanded RNAscope analysis, the authors conclude that "the combination of fluorescence in situ hybridization with IHC and tissue clearing allowed the confirmation of a small subset of the Trailblazer transcriptional signature". This statement seems to imply that heterogeneous expression is sufficient to validate transcripts as being part of the Trailblazer signature. According to the Introduction of the manuscript such genes should be expressed by cells "narrowly confined by the migratory front". This discrepancy needs to be addressed or the reader will not be able to link the transcriptomic data with the in vivo expression analysis.

We completely agree with the reviewers and have revised both the Abstract, Introduction and Results to clarify our results and establish a better link between the transcriptomic data and in vivo expression analysis.